# Unveiling the critical role of androgen receptor signaling in avian sexual development

Kamila Lengyel[1,2,3], Mekhla Rudra[1,3], Tom V. L. Berghof [2], Albertine Leitão[1], Carolina Frankl-Vilches[1], Falk Dittrich[1], Denise Duda[2], Romina Klinger[2], Sabrina Schleibinger[2], Hicham Sid [2], Lisa Trost[1], Hanna Vikkula[2], Benjamin Schusser [2,4] & Manfred Gahr [1,4] ✉

Gonadal hormone activities mediated by androgen and estrogen receptors, along with cell-autonomous mechanisms arising from the absence of sex-chromosome dosage compensation, are key factors in avian sexual development. In this study, we generate androgen receptor (AR) knockout chickens (AR$^{-/-}$) to explore the role of androgen signaling in avian sexual development. Despite developing sex-typical gonads and gonadal hormone production, AR$^{-/-}$ males and females are infertile. While few somatic sex-specific traits persist (body size, spurs, and tail feathers), crucial sexual attributes such as comb, wattles and sexual behaviors remain underdeveloped in both sexes. Testosterone treatment of young AR$^{-/-}$ males fails to induce crow behavior, comb development, or regression of the bursa of Fabricius, which are testosterone-dependent phenotypes. These findings highlight the significance of androgen receptor mechanisms in fertility and sex-specific traits in chickens, challenging the concept of a default sex in birds and emphasizing the dominance of androgen signaling in avian sexual development.

Secondary sexual characteristics and reproductive behaviors in vertebrates have traditionally been attributed to the influence of gonadal sex determination, where testicles or ovaries largely dictate the development of these traits through the actions of androgens and estrogens[1]. However, recent evidence challenges the conventional paradigm, suggesting that sex chromosomal activity in the brain directly influences sex-specific differences in mammals and birds[1]. Avian sexual development, in particular, involves brain autonomous mechanisms in shaping sexual behaviors[2,3], suggesting a more intricate regulation beyond gonadal hormones alone. Unlike mammals, avian cells possess cell autonomous sex identity, allowing somatic cells to develop sexual traits independent of hormonal influence due to the absence of sex chromosome dosage compensation[4,5]. This unique characteristic has significant implications, potentially leading to the manifestation of sexual traits differing from gonadal sex expectations. Remarkably, male chickens with experimentally reversed gonads develop complete sexual ornaments typical of roosters[6], challenging the notion that these traits are driven by testicular hormones. Our study aims to unravel the intricate interplay between hormonal and cell autonomous mechanisms in avian sexual development by investigating the role of androgen signaling through androgen receptor knockout in chickens. We seek to shed light on the contributions of androgen signaling to the fertility and sexual characteristics of male and female birds.

Androgens and estrogens exert their effects by binding to transcription factors, namely the androgen receptor (AR) or estrogen receptors, leading to direct changes in gene expression[7]. In mammals, ARs are widely expressed in various tissues[8], and testosterone plays a

[1]Department of Behavioural Neurobiology, Max Planck Institute for Biological Intelligence, Seewiesen, Germany. [2]Reproductive Biotechnology, TUM School of Life Sciences Weihenstephan, Technical University of Munich, Freising, Germany. [3]These authors contributed equally: Kamila Lengyel, Mekhla Rudra. [4]These authors jointly supervised this work: Benjamin Schusser, Manfred Gahr. ✉e-mail: manfred.gahr@bi.mpg.de

crucial role in the development of male traits, encompassing both secondary sexual characteristics[8–10] and behavior[11–13]. Disruptions in AR signaling in humans or mice result in exclusive development of secondary female phenotypes[14,15], as observed in individuals with spontaneous AR mutations and studies using AR knockout mouse models[9,16–19]. However, it remains unclear whether disruptions in the avian androgen signaling system yield comparable phenotypic outcomes due to developmental differences between mammals and birds.

The effects of gonadal steroid hormones in birds have been extensively investigated using methods such as castration, gonad transplants and hormone administration[20–22]. These studies have demonstrated the influence of testosterone and estrogen on brain and behavioral development in galliform birds and songbirds[22–24]. However, the precise role of androgen signaling in avian sexual development has remained elusive due to the lack of specific experimental avian models. Recent advancements in cultivating and editing primordial germ cells have allowed for the generation of precise bird models[6,25–29]. In this study, we introduce a novel chicken line with AR knockout (AR$^{-/-}$), providing valuable insights into avian sexual development. We aim to elucidate the role of androgen signaling in avian sexual development by investigating the fertility and differentiation of secondary sexual characteristics in AR$^{-/-}$ roosters and AR$^{-/-}$ hens. Surprisingly, we found that disruption of AR/androgen signaling significantly affects both male and female phenotypes, influencing secondary sexual ornaments, behaviors and reproduction despite continued production of gonadal steroid hormones. Through this avian model, we delve into the mechanisms governed by androgen signaling in avian sexual differentiation, to gain insights in both the similarities and differences between avian and mammalian sexual phenotypes.

## Results and discussion
### Generation of the androgen receptor (AR) knockout (AR$^{-/-}$) chicken
We used CRISPR/Cas9 technology with homologous directed repair, to delete exon 2, which contains the DNA binding domain of the chicken androgen receptor (AR) (Supplementary Fig. S1). We successfully generated seven clonal homozygous AR$^{-/-}$ primordial gem cell lines, and among them, two lines were used to produce a total of 17 germline chimeras. PCR analysis of sperm-derived genomic DNA (gDNA) confirmed germline transmission in ten out of 17 chimeras. Two of these chimeras were bred with wild-type AR$^{+/+}$ hens, resulting in the production of heterozygous AR$^{+/-}$ offspring. The germline transmission rates were determined to be 2.6% for chimera 1 and 0% for chimera 2 (Supplementary Table S1). In homozygous AR$^{-/-}$ chicken, our analysis revealed the absence of AR mRNA including exon 2 and exon 3 in embryonic gonads (Supplementary Fig. S2, PCR analysis) and the absence of peptides related to exons 2 to 8 in adult testes (Supplementary Table S2; LC-MS/MS analysis). This confirms that the loop-out of exon 2 (Supplementary Fig. S1) resulted in a truncated AR protein, which included only exon 1 and, therefore, was non-functional, lacking the DNA binding domain (exon 2) and the hormone binding domain (exon 3).

### Sexual phenotypes of AR$^{-/-}$ male and AR$^{-/-}$ female chicken
Heterozygous AR$^{+/-}$ males and females (Supplementary Fig. S3) exhibited no noticeable phenotypical differences compared to AR$^{+/+}$ chickens (Fig. 1b, d) and both were fertile. There were no observable variations in egg-laying behavior (Fig. 2f) in the AR$^{+/-}$ females as compared to the AR$^{+/+}$ females, which both differed significantly from the egg-laying of the AR$^{-/-}$ females (one-way ANOVA followed by Tukey-HSD: (F(2) = 193.34, $p < 0.00001$). These findings strongly suggest that the presence of a single functional copy of the AR is sufficient to maintain fertility, sexual characteristics, and behaviors in chickens.

In stark contrast, AR$^{-/-}$ males and females displayed a distinct phenotype characterized by infertility and the absence of typical secondary sexual characteristics (Fig. 1). Disruption of the AR led to the lack of head sexual ornaments, including combs, wattles, earlaps, and eye ring pigmentation in both AR$^{-/-}$ roosters and AR$^{-/-}$ hens. For instance, AR$^{-/-}$ roosters developed only very small combs (Fig. 1a); likewise, AR$^{-/-}$ hens did not develop typical female combs (Fig. 1c) compared to AR$^{+/+}$ (Fig. 1b, d). At 20 weeks of age, when animals reached adulthood, the weights of combs of AR$^{+/+}$ roosters (42.5 ± 8.2 g, mean ± sd) were significantly bigger than those in AR$^{-/-}$ roosters (0.30 ± 0.06 g) (t(6) = 8.9, $p = 0.0045$, one-tailed t-test; Supplementary Table S5a). Similarly, the weights of combs in AR$^{+/+}$ hens (7.5 ± 3.8 g) significantly differed from those in AR$^{-/-}$ hens (0.16 ± 0.04 g); (t(7) = 3.79, $p = 0.003$, one-tailed t-test; Supplementary Table S5a). Interestingly, the comb weights of AR$^{-/-}$ roosters and AR$^{-/-}$ hens remained sexually dimorphic (t(5) = 3.95, $p = 0.005$, one-tailed t-test; Supplementary Table S5a). However, other external secondary sexual characteristics such as tail feathers, spurs, body size and weight remained intact (Fig. 1a, c). AR$^{-/-}$ chicken remained sexually dimorphic, with larger males, and identical weight of AR$^{-/-}$ and AR$^{+/+}$ roosters (Fig. 1a1, b1 versus Fig. 1c1, d1) (AR$^{-/-}$: 1.54 ± 0.12 kg; AR$^{+/+}$: 1.69 ± 0.19 kg; (t(5) = 1.61, $p = 0.120$, one-tailed t-test; Supplementary Table S5a). This is unexpected considering that androgens are known to decrease the body weight of chickens[30], which is the primary reason for castrating male chickens in commercial capon production.

### AR$^{-/-}$ males develop testicles and AR$^{-/-}$ females develop ovaries but both are infertile
Gonad determination remained unaffected in the AR$^{-/-}$ chickens, with AR$^{-/-}$ males developing testicles and AR$^{-/-}$ females developing ovaries (Fig. 1a3, c3). Among adult males, however, it was notable that AR$^{-/-}$ roosters had smaller testicles compared to their AR$^{+/+}$ counterparts (Fig. 1a3, b3), as evidenced by the significantly lower testicles-to-body weight ratio (Fig. 2e; one-tailed t-test, t(4) = 14.9, $p = 0.00006$; Supplementary Table S5a). Histological analysis revealed that the AR$^{-/-}$ testicles deviated from the typical organization found in AR$^{+/+}$. In particular, seminiferous tubules had smaller diameter due to a reduced or absent lumen, and they were more randomly distributed across the testicles, accompanied by massively expanded interstitial tissue (Fig. 1a4, b4). Specifically, we observed a significantly smaller diameter of the seminiferous tubules (AR$^{+/+}$: 452 ± 49 μm; AR$^{-/-}$: 211 ± 55 μm; t(4) = 5.67, $p = 0.002$, one tailed t-test, Fig. 1). The proportion of interstitial tissue was 9.8 ± 3.7% in AR$^{+/+}$ testicles and 60.7 ± 8.9% in AR$^{-/-}$ testicles (see Supplementary Fig. S4). The expanded interstitial tissue is composed of Leydig cell-like cells expressing luteinizing hormone (LH) receptor (LHR) mRNA and of interstitial cells not expressing LHR mRNA (Supplementary Fig. S4). The much higher expression of LHR receptors in Leydig cells of the AR$^{-/-}$ roosters compared to AR$^{+/+}$ roosters (AR$^{+/+}$: 1.5 ± 0.16 dots per cell; AR$^{-/-}$: 4.46 ± 0.39 dots per cell; t(4) = 12.24, $p = 0.0001$, one tailed t-test; Supplementary Fig. S4) as well as the hyperplasia of these cells suggests permanent high levels of LH, as found in mammalian species[31]. However, plasma LH levels were barely higher in AR$^{-/-}$ than in AR$^{+/+}$ roosters (two-tailed t-test, t(24) = 2.086, $p = 0.048$, see Supplementary Table S7) reminiscent of rare cases of Leydig cell hyperplasia and low LH levels in AR-deficient humans[32].

Complete infertility was evident, as no mature sperms were found in AR$^{-/-}$ roosters while the distribution of Sertoli cells and spermatogonia was maintained in their testis (Fig. 1a4, 1b4; Supplementary Figs. S2a, b, S5). While spermatogonia and primary spermatocytes were abundant in AR$^{-/-}$ seminiferous tubule, the formation of spermatids was markedly limited, observed in minimal quantities in only about 5% of the tubules (insert in Fig. 1a4, Supplementary Fig. S5a) compared to AR$^{+/+}$ tubules (insert in Fig. 1b4, Supplementary Fig. S5b). AR mRNA was detected in Leydig cells, Sertoli cells, spermatogonia and round spermatids of wild-type roosters (Fig. S6). Sertoli cells are

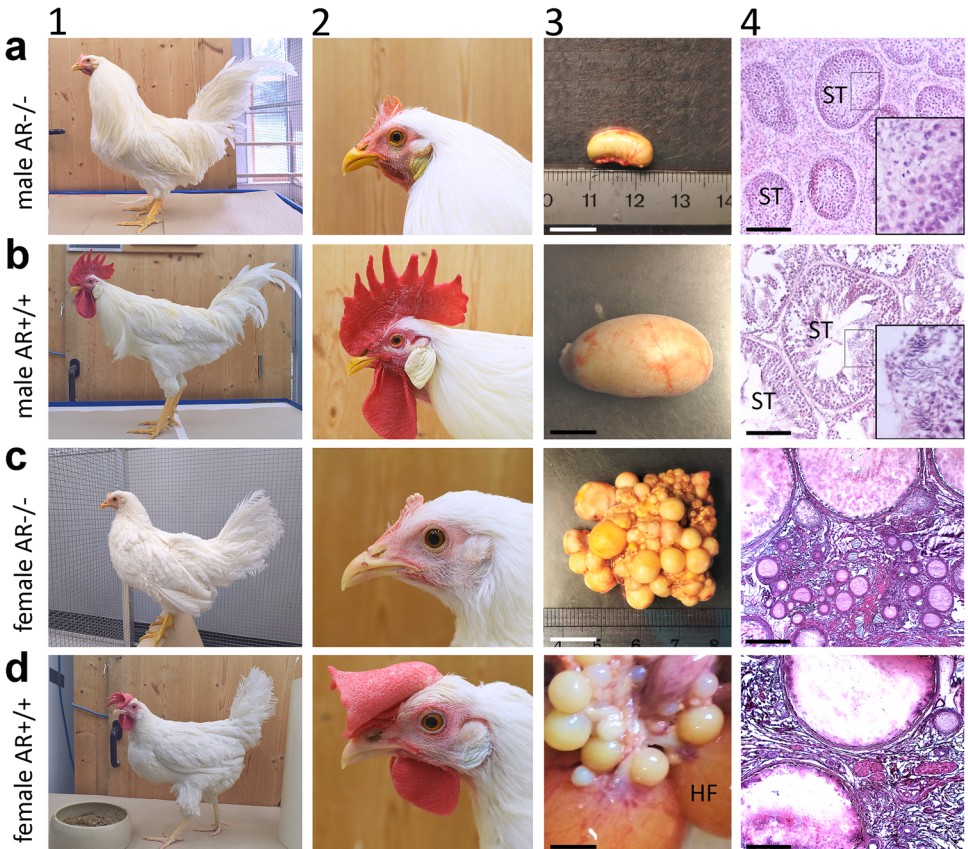

**Fig. 1 | Male and female chickens with complete androgen receptor knockouts lack male and female sexual ornaments, respectively, and are infertile due to impaired gametogenesis.** a–d Depict the phenotypic differences observed between homozygous AR⁻/⁻ males (**a1, a2**) and AR⁻/⁻ females (**c1, c2**) compared to AR⁺/⁺ males (**b1, b2**) and AR⁺/⁺ females (**d1, d2**) at adulthood, being 20 weeks old. The AR⁻/⁻ males and females lacked typical sexual characteristics such as comb, earlaps and wattles. However, there were notable differences in body size, plumage (presence of sickle tail feathers in males), and leg ornaments (presence of spurs in males) between AR⁻/⁻ males and females. "3" depicts the differences of the gonads and "4" their morphology (Haematoxylin stained paraffin (**a4, b4**) and cryostat (**c4, d4**) sections). The testicles of AR⁻/⁻ males (**a3**) were significantly smaller compared to those of AR⁺/⁺ roosters (**b3**) (see Fig. 2e). Morphologically, AR⁻/⁻ testicles lacked

the typical organization, especially the seminiferous tubules (ST, arrows) with lumen (**a4**) compared to AR⁺/⁺ testicles (**b4**). The mean diameters of tubules were 493, 464 and 398 μm for three AR⁺/⁺ and were 268, 208 and 158 μm for three AR⁺/⁺ males, respectively. In addition, the seminiferous tubules of the AR⁻/⁻ testicles lacked mature sperm (insert in A4) compared to the AR⁺/⁺ testicles (insert in **b4**). In **a4** and **b4**, the small squares indicate the enlarged areas of the inserts. The ovaries of AR⁻/⁻ hens contained smaller follicles of varying sizes (**c3, c4**) but lacked the late-stage hierarchical follicles (HF) seen in AR⁺/⁺ hens (**d3, d4**), and they lacked the ovulated oocytes (not depicted). Histological staining was performed for gonads of three individuals per group. Scale bar represents 1 cm in **a3–d3** and 200 μm in **a4–d4**, and represents 50 μm for the large inserts in **a4** and **b4**.

known to be sensitive to testosterone and play a crucial role in creating the microenvironment for spermatogenesis through secreted signals[33]. The PGCs used to generate the knockout line had an AR⁻/⁻ genotype and were injected into an AR⁺/⁺ gonadal environment of the recipient rooster, resulting in the development of functional AR⁻/⁻ spermatozoa. This underlines that AR expression in the germ cell line does not play a role in spermatogenesis, but that AR expression of Sertoli cells is crucial for the induction of signaling supporting spermatogenesis in the chicken. In particular, AR appears to be relevant for the transition from primary spermatocytes to spermatids. In mouse models with Sertoli cell-specific AR ablation, meiosis and thus spermatogenesis are completely blocked[34]. The observed residual spermatogenesis activity in AR⁻/⁻ chickens could be due to high follicle stimulating hormone (FSH) levels, which induce some of the otherwise AR-dependent mechanisms of Sertoli cells[35]. Indeed, FSH plasma levels were significantly elevated in AR⁻/⁻ males as compared to AR⁺/⁺ males (two-tailed t-test, t(23) = 2.252, $p$ = 0.019; Supplementary Table S7).

Among adult females, AR⁻/⁻ females had smaller ovaries than AR⁺/⁺ and AR⁺/⁻ females due to the absence of late-stage (so called hierarchical) follicles and ovulated oocytes (Fig. 1c3, c4, d3, d4; Supplementary Fig. S3), and they did not lay eggs (Fig. 2f; Supplementary

Table S5b). We note that, akin to the testes, the germ cell markers VASA and DAZL[36,37] were also present in the ovaries of AR⁻/⁻ females (Supplementary Fig. S2a, b). ARs are prominently expressed in the granulosa cells of pre-ovulatory follicles and in thecal interstitial cells (ref. 38; Supplementary Fig. S6). While AR is expressed in the ovary during embryonic age[39], our data suggests that androgen signaling may not play a crucial role in the organization of the ovary until puberty. Thus, the arrest of follicle development in AR⁻/⁻ ovaries may be attributed to the lack of functional ARs in these follicles, since ARs are important to initiate the expression of bird-specific regulatory pathways to unable follicular transition to the preovulatory stage[40]. In addition, compared to AR⁺/⁻ and AR⁺/⁺ hens, these hens had significantly smaller oviducts (not shown) and proliferation of the uterine part of the oviduct, known to be sensitive to testosterone and to contain ARs[41,42], was absent. Therefore, oviduct functioning was likely compromised, and resulted in an unsupportive environment in the unlikely event of ovulation. These factors contributed to the failure of ovulation and egg laying in AR⁻/⁻ hens. In relation to the disrupted fertility, FSH plasma levels were significantly elevated in AR⁻/⁻ females as compared to AR⁺/⁺ females (two-tailed t-test, t(15) = 5.512, $p$ = 0.00006; Supplementary Table S7).

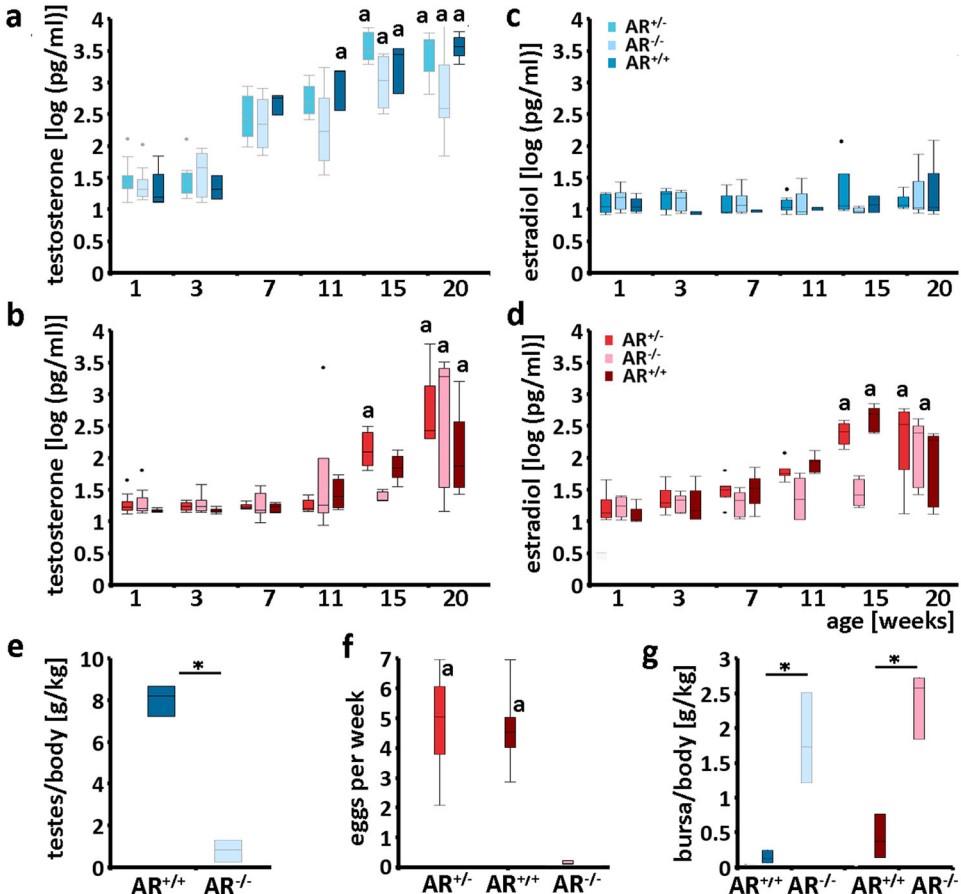

**Fig. 2 | Testosterone and estradiol plasma profiles maturate similarly in wild-type (AR$^{+/+}$), heterozygous (AR$^{+/-}$), and homozygous (AR$^{-/-}$) juveniles while development of testicle size, egg-laying and weight of bursa of Fabricius is impacted in AR$^{-/-}$ chickens.** We depict plasma levels (pg/ml) in males and females and from 1 to 20 weeks after hatching (for all groups $n \geq 3$, Supplementary Table S3 or Source Data file), corresponding to sexual maturity. At 20 weeks, testosterone levels significantly increased as in all male (**a**) and all female (**b**) genotypes compared to weeks 1 to 11. Estradiol levels also increased significantly during puberty in females but not in males (**c**, **d**). For **a–d**: Standard Least Squares (LS) and incorporated Restricted Maximum Likelihood estimation followed by LS Means Tukey HSD post hoc tests for pairwise comparisons; see Supplementary Table S4. Similar letters in a-d indicate that these groups were statistically similar and different from all other groups (see text and Supplementary Table S4 for all statistical results). In **e**, the graph shows the testicle-weight to body-weight ratio at 20 weeks of AR$^{-/-}$ males (dark blue, $n = 3$) to be significantly lower compared to AR$^{+/+}$ males (light blue, $n = 3$) (one-tailed t-test, t(4) = 14.9, $p = 0.00006$). **f** While AR$^{+/+}$ (dark red) and AR$^{+/-}$ (red) females laid approximately 5 eggs per week ($n = 4$ females per group), no eggs were laid by AR$^{-/-}$ hens (light red) (one-way ANOVA followed by Tukey-HSD: (F(2) = 193.34, $p < 0.00001$; Supplementary Table S5b or Source Data file). Similar letters in **f** indicate that these groups were statistically similar. Panel **g** shows at 20 weeks the bursa of Fabricius-weight to body-weight ratio of males (blue bars) and females (red bars) of AR$^{-/-}$ and AR$^{+/+}$ individuals. The bursa weight was higher in AR$^{-/-}$ compared to AR$^{+/+}$ in both sexes ($n = 3$ per group; one-tailed t-tests, females: t(4) = 6.063, $p = 0.002$; males: t(4) = 6.537, $p = 0.001$; Supplementary Table S5a or Source Data file). For **e**, **g***: $p < 0.05$ (see above). For **a–d** and **f**: Shown are box plots with the median (line inside the box), first and third quartiles and outliers. Whiskers are the smallest and largest values within 1.5 times the interquartile range. In **e** and **g**, the box and inner line represent the three data points.

---

In conclusion, AR-androgen signaling plays a crucial role in spermatogenesis in male and the differentiation of late-stage follicles, oocytes and eggs in female chicken. An AR-sensitive signaling mechanism of gametogenesis concerns the components of the activin/inhibin signaling pathway that regulates FSH production in the pituitary gland[43], which could lead to elevated FSH levels in the AR$^{-/-}$ chicken. Since testosterone levels are similar in AR$^{-/-}$ and wildtype chickens (see next paragraph), the lack of a functional AR in gametogenesis cannot be compensated for by AR independent testosterone driven or non-steroidal mechanisms. In mammals, AR mutant males are infertile and females with ovarian knockouts of the AR are subfertile[17–19].

**Testosterone and estradiol profiles of AR$^{-/-}$ male and female chicken during development**
Next, we examined hormone production, by monitoring testosterone and 17ß-estradiol plasma levels in males and females from hatching until sexual maturity at 20 weeks of age (Fig. 2a–d; Supplementary Table S3). The overall patterns of testosterone plasma levels during ontogeny were similar between AR$^{+/+}$, AR$^{+/-}$, and AR$^{-/-}$ males, showing a substantial increase after 3 weeks and again at 15–20 weeks of age, corresponding to the transition from puberty to adulthood (Fig. 2a; Supplementary Table S3, for statistics [Restricted Maximum Likelihood estimation] see Supplementary Table S4). Interestingly, like the males, all female genotypes showed a similar increase in testosterone levels at 15–20 weeks (Fig. 2b; Supplementary Tables S3, S4). In contrast, estradiol levels remained low in all males but increased in all females around 15 to 20 weeks of age (Fig. 2c, d; Supplementary Tables S3, S4). These observations indicate that gonadal steroid production in males and females from all groups rises from puberty to adulthood, suggesting the presence of a functional hypothalamus-pituitary-gonad. This functionality does not appear to be mediated by the AR or by feedback mechanisms with gonadal estrogens, as testosterone production increases while plasma estrogen levels remain constant throughout puberty in males (Fig. 2a, c). If we assume that chicken Leydig cells' steroidogenesis is LH- and AR-regulated as in

mammals[44], we expect the functioning hypothalamic-pituitary-gonadal axis should lead to high LH release due to initially low testosterone production[31]. LH levels were just about, but significantly, higher in AR[−/−] than in AR[+/+] males (see above; Supplementary Table S7) and testosterone plasma levels (Fig. 2a) were similar between AR[−/−] and AR[+/+] males, while we observed high expression of LHR mRNA in Leydig cells and substantial proliferation of LHR mRNA-expressing cells in the testicular interstitium (Supplementary Fig. S4). These data suggest that each LHR-positive cell of AR[−/−] birds constitutively produces low testosterone levels, leading to normal elevated testosterone levels in adult AR[−/−] males due to the sheer number of these cells. Thus, the hypothalamic-pituitary-gonadal axis of AR[−/−] males actually functions through a different mechanism (hyperplasia of steroidogenic testicular cells) compared to AR[+/+] roosters (regulation of LH production). In AR mutant humans, too, testosterone production is rather normal[31]. What then triggers the proliferation of LHR expressing cells in AR[−/−] roosters needs to be seen. Future, detailed analysis of the expression of all major components of the hypothalamic-pituitary gonadal axis of male and female AR[−/−] and AR[+/+] chicken are needed to better understand how the steroidogenesis could be maintained while the gametogenesis is disrupted.

Further, since the cell lineages that give rise to the Sertoli and Leydig cells of the testis and the cell lineages that give rise to the theca and granulosa cells of the ovary differ between mammals and chickens[45], the regulatory mechanisms, including androgen-dependent ones, in these cell types involved in steroidogenesis and gametogenesis may differ between chickens and mammals. Thus, the expression of LHR and the number of Leydig cells in the mouse are AR-dependent[44,46]. Nevertheless, since the testosterone and estradiol plasma levels of AR mutants are comparable to those of intact animals, the observed phenotypes of male and female AR[−/−] are due to the lack of transcriptional activity of the ARs, but neither to insufficient androgen production and its metabolites nor to non-genomic effects of testosterone[47].

## AR dependent development of the Bursa of Fabricius, the comb and the crowing

We experimentally investigated three well-established androgen-sensitive phenotypes: bursa of Fabricius involution, rooster crowing and comb development. The bursa, an immune organ in birds, naturally undergoes regression, so-called involution, during puberty between approximately 15 to 20 weeks of age in the chicken. This process is thought to be driven by rising testosterone levels, with bursae regressing faster in males than in females[48]. Remarkably, we observed significant delay in bursal involution in both AR[−/−] males and females at this age compared to AR[+/+] individuals, as indicated by bursa-weight to body-weight ratio, and in some cases, regression was not initiated at all (one-tailed t-tests, females: t(4) = 6.063, p = 0.002; males: t(4) = 6.537, p = 0.001; Fig. 2g, Supplementary Fig. S7, Supplementary Table S5a). ARs identified by AR mRNA expression are mainly located in the mesenchymal cells of the plicae epithelium, in the laminae surrounding the follicles and in the follicular medulla in the bursa (Supplementary Fig. S7). Likely, these cells are the sites for the AR-dependent triggering of cell death underlying the regression of the bursa in the AR[+/+] chicken.

Artificial induction of bursa involution in both sexes during the early embryonic stage may be due to testosterone treatment, although the exact underlying mechanisms are not clear and estrogen and progestin activity are also thought to be involved[49]. Interestingly, our experiments demonstrated that exposure of AR[−/−] embryos of both sexes (egg dipping) to testosterone failed to induce regression of the bursal morphology, unlike AR[+/+] individuals (Fig. 3). Untreated embryos showed the typical structure of the bursa, with plicae subdivided in lymphoid follicles filled with B lymphocytes (Fig. 3c, d). In AR[+/+] embryos treated with testosterone, the follicle density (and thus

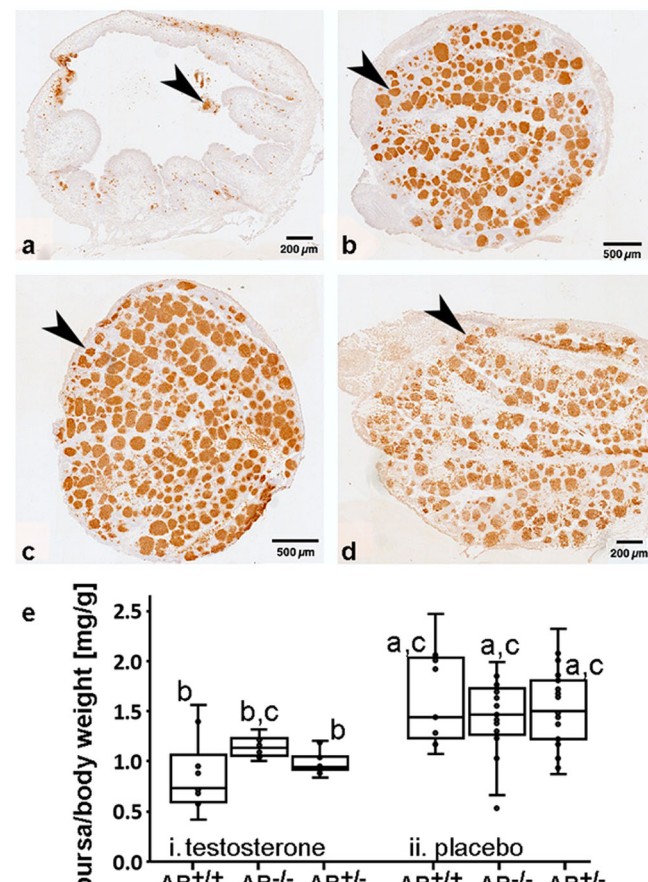

**Fig. 3 | Testosterone induced regression (involution) of the bursa of Fabricius in wild-type (AR[+/+]) but not in homozygous (AR[−/−]) embryos. a–d** Sections of the bursa immunolabeled for B-lymphocytes, with brown labeled follicles (arrowheads) indicating the presence of lymphoid tissue at embryonic day 18. In the testosterone-treated AR[+/+] embryos, the bursa (**a**) lacked plicae and had a larger lumen, suggesting regression of the organ while the bursa of untreated AR[+/+] embryos showed normal histology (**c**). Further, remaining plicae of testosterone-treated AR[+/+] embryos were widely depleted of lymphoid follicles (**a**). In contrast, testosterone treatment had no effect on the bursal organization in AR[−/−] embryos (**b**) compared to untreated AR[−/−] embryos (**d**), as the plicae contained numerous follicles in both cases. For size comparison scale bars representing 200 μm (**a**, **d**) and 500 μm (**b**, **c**), respectively, are given. Histology was performed in three animals per group. The ratios of the bursa weight to body weight (mg/g) (**e**), demonstrate a significant effect of testosterone in AR[+/+] and AR[+/−] embryos compared to placebo treated groups while this was not the case for the testosterone treated AR[−/−] embryos. For each group, male and female embryos were combined for the statistical analysis using one-way ANOVA followed by Tukey-HSD, F(5) = 12.50, p < 0.01; groups labeled with different letters are statistically different; n ≥ 8 for all groups). Shown are box plots with the median (line inside the box), first and third quartiles. Whiskers are the smallest and largest values within 1.5 times the interquartile range. For the data, see Supplementary Table S6 or the Source Data file.

the number of lymphocytes) in the bursa was greatly reduced (Fig. 3a). In contrast, the morphology and lymphoid follicle density of the bursa of AR[−/−] embryos remained unaffected by testosterone treatment (Fig. 3b). Further, the weight of the bursa per gram body weight of testosterone treated AR[−/−] animals was similar to that placebo treated animals while that of AR[+/+] and AR[+/−] embryos was significantly reduced (Fig. 3e; for each group (n ≥ 8), male and female embryos were combined for the statistical analysis using one-way ANOVA followed by Tukey-HSD, F(5) = 12.50, p < 0.01; see Supplementary Table S6 or Source Data file). The results not only affirm the successful elimination of the AR but also indicate the significance of AR-regulated immune

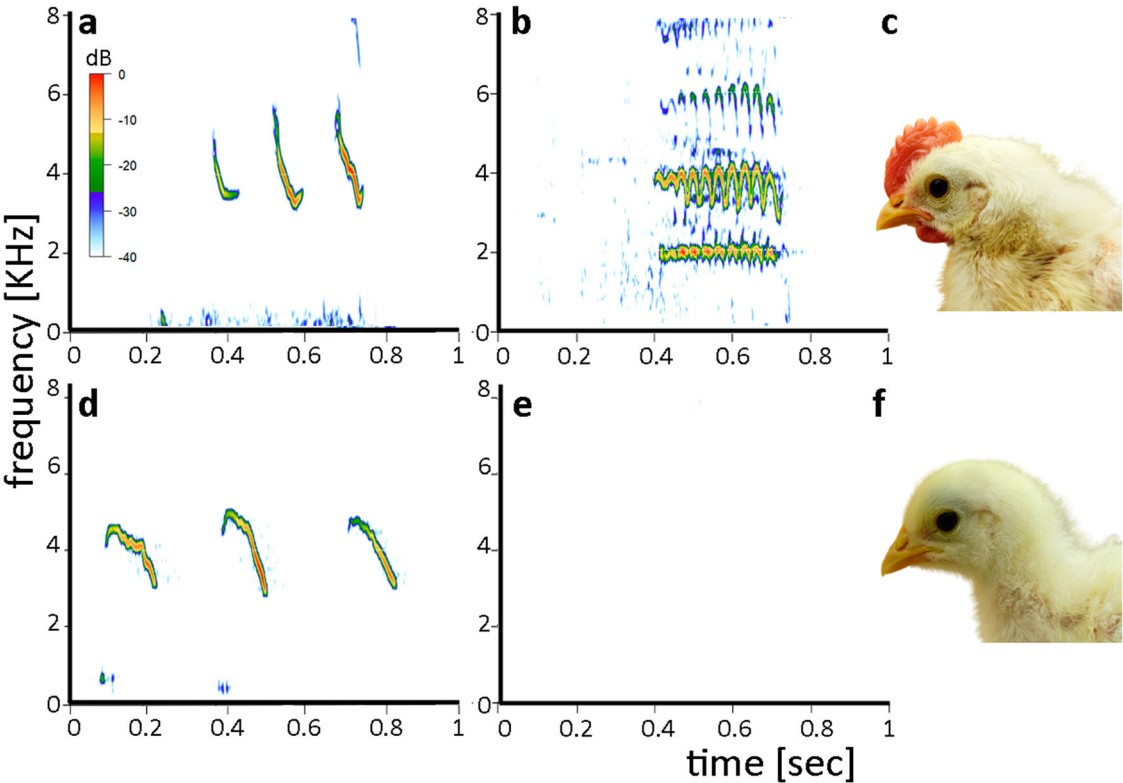

**Fig. 4 | Testosterone treatment stimulate the early development of secondary sexual characteristics (crowing, comb) in heterozygous (AR$^{+/-}$) male chicks but not in homozygous (AR$^{-/-}$) ones.** In AR$^{+/-}$ males (upper panels), testosterone treatment induced precocious crowing (high-pitched vocal trills) (**b**) and comb growth (**c**), whereas in AR$^{-/-}$ males (lower panels), these sexual phenotypes did not develop (**e**, **f**). Chicks from both groups display distress calls that show individual variations, which were not affected by testosterone treatment (**a**, **d**). This suggests that the development of sexual phenotypes depends on the presence of at least one functional allele of the androgen receptor (AR) gene. Testosterone treatment was through a subcutaneous hormone depot implanted at seven days post-hatching. Chicks began to crow in the first week after testosterone treatment, while comb development started in the second week. The sound amplitude (dB) is color coded.

system development in avian sexual differentiation – an aspect that has been overlooked in recent decades.

Crowing behavior can be induced through testosterone treatment in male chicks[20]. Following testosterone implantation at 7 days post-hatching, AR$^{+/-}$ male chicks began to crow frequently at 12–16 days, while the testosterone implanted AR$^{-/-}$ male chicks did not show this behavior ($n = 3$ for each group) (Fig. 4). This suggests that the induction of crowing behavior by testosterone depends on neural circuits that are sensitive to AR activation and fail to fully differentiate in the absence of functional ARs. The hindbrain syringeal motor nucleus and specific midbrain regions, known to regulate species-typical crow patterns[50,51], are likely androgenic sites that express ARs[52,53]. If estrogen signaling is able to replace in part androgen signaling for the control of vocalizations needs to be seen. At 20 weeks, next to typical chicken social calls, adult AR$^{-/-}$ roosters occasionally produce vocalizations that do not fit typical male or female patterns. These unique vocalizations shall be subject to detailed analysis in future studies.

Furthermore, testosterone treatment in male chicks led to the development of sexual head ornaments in addition to the developing crow. In the testosterone-treated AR$^{+/-}$ chicks, the development of a comb began at about 13–20 days of age while the testosterone-treated AR$^{-/-}$ chicks did not develop a comb (Fig. 4c, f), thereby confirming the AR dependence of this sexual ornament. The stratum germinativum of the epidermis of the comb expresses ARs[54].

### AR determines most sexual phenotypes of the chicken
As demonstrated in both AR$^{-/-}$ males and females, the AR plays a crucial role in genetic mechanisms governing fertility and the development of secondary sexual exo- and endophenotypes, including sexual behaviors. While estrogen receptors may contribute to avian sexual development of both sexes[23,55,56], estrogenic mechanisms alone are insufficient for the emergence of most sexual phenotypes, including fertility and reproductive behaviors. Instead, AR signaling is necessary in male and female chicken. Further investigation is needed to determine the degree to which persistent secondary sex characteristics, the body size, spurs, tail feathers in AR$^{-/-}$ individuals, and elements of socio-sexual behaviors depend on estrogenic mechanisms or develop through steroid hormone-independent cell-autonomous mechanisms. In particular, the prevailing notion of estrogen-induced demasculinization of male behaviors during avian ontogeny[56–59], should be reevaluated in light of the strong dependence of male and female sexual phenotypes on functional AR. This "estrogen" hypothesis relied on the administration of excessively high concentrations of estrogen to embryos and hatchlings[57–60], resulting in high mortality rates[57], and only partially explored the possibility of gonadal regression and reduced testosterone production[61–63]. The demasculinized phenotype of estrogen-treated avian embryos could be attributed to various factors, such as pharmacological disruption of testosterone production in the gonad, downregulation of gonadotropin signaling to the gonad due to the negative feedback of estrogens in the hypothalamus, or a cell type-specific alteration of AR expression in the brain. In relation to this, infusion of testosterone into the hypothalamus of adult cocks hormonally demasculinized during the embryonic period triggered mating behavior[64]. Further, evidence from spontaneous behavioral sex reversals of adult birds[65], testosterone-induction of male behaviors in pubertal female chickens[57] and adult female quails[66,67], as well as observations from transgenic roosters overexpressing aromatase throughout their lives and displaying male sexual behaviors despite

elevated estrogen levels[68], all contradict the concept of estrogen-driven demasculinization. Given the widespread expression of ARs in most tissues of both male and female birds, we propose that androgens directly influence diverse tissues, leading to sex-specific phenotypes. In mammals, mutations in ARs have also been found to have pronounced phenotypic effects on gonadal functions and sexual behaviors in both sexes[69,70]. These observations strongly suggest that androgens play a crucial role in shaping sexual development and behavior in birds and mammals. In both birds and mammals, mutations that directly or indirectly affect cell type-specific expression of ARs would impair sexual development and behavior.

The absence of functional ARs resulted in a partial demasculinization of roosters and a partial defeminization of hens, challenging the concept of a default sex of male and female genomes. Specifically, males without ARs do not develop the overall female phenotype, and females without ARs do not develop the overall male phenotype. In relation to this, gynandromorphic birds display distinct male and female characteristics on opposite halves of their bodies[5]. Such chickens with a predominantly male-typical wattle on one side and a predominantly female-typical wattle on the other exemplify the androgenic activation of body-half-specific genetic mechanisms. Either the androgen-induced cell proliferation or the AR abundance could be different between male and female wattle tissue in gynandromorphic birds. As shown in the present study, sexual dimorphism persists in head ornamentation, such as the combs of AR[−/−] males and females (Fig. 1; Supplementary Table S5a), supporting the assumption that the tissues respond to AR signaling in a sex-specific, tissue-autonomous manner.

The underlying mechanisms responsible for the development of remaining sexual traits in AR[−/−] chicken, whether influenced by estrogenic factors or reliant on cell-autonomous sex-chromosome-based signaling, require further investigation. To discern the relative contributions of androgenic, estrogenic and cell-autonomous mechanisms of sexual phenotypes, an effective approach would be the local inactivation of estrogen receptors and aromatase in tissues of AR[−/−] males and females. Nevertheless, it is clear that AR-dependent mechanisms extend beyond modulating the degree of sexual differentiation as they serve as the primary determinant for many sex-specific functions in male and female birds. However, given that ARs rely on androgens, a plausible explanation for both AR-driven effects observed in this study and brain-autonomous and cell-autonomous sexual development described in previous research[2–6] is that the latter might determine the properties of the hypothalamus that control the sex-specific production of gonadal hormone levels.

## Methods

### Animals

The Lohmann's Selected Leghorn Classic (LSL) chicken line, obtained from LSL Rhein-Main (Dieburg, Germany), was used in this study. These chickens were housed under conventional breeding conditions and provided with ad libitum access to food and water. They were accommodated in the S1 animal facility at the Technical University Munich (TUM Animal Research Center, Weihenstephan Technical University of Munich) and the S1 animal facility at the Max Planck Institute for Biological Intelligence, Seewiesen, Germany. All experimental procedures related to the generation of AR knockout chickens were approved by the government of Upper Bavaria under the license number ROB 55.2.2532.Vet.02-17-101, while the procedures involving testosterone implantation and blood sampling were conducted with approval under the license number ROB-55.2-2532.Vet_02-20-126. At three weeks or twenty weeks of age the animals were euthanized using an overdose of isoflurane. Tissues including the gonads and bursae of Fabricius were collected, snap-frozen on dry-ice, and stored at −80 °C for further analysis. Data are available in the Supplementary

Tables S1–S7, included in the Supplementary Materials. The AR knockout chicken line is available upon request.

### Generation of AR knockout chickens

To generate AR[+/−] and AR[−/−] chickens, we designed a specific guide RNA (sgRNA) with the sequence 5′- CAAAGTGTTCTTCAAGCGGG − 3′ (Fig. S1B; https://benchling.com/crispr) targeting exon 2, which contains the DNA binding domain of the chicken AR (Fig. S1A). The sgRNA was cloned into the pX330-U6-Chimeric_BB-CBh-hSpCas9 (#42230) vector obtained from Addgene, USA, using the BbSI restriction enzyme, following established protocols[71]. The targeting vector for homologs directed repair contained a 5′ and a 3′ homology region flanking a selectable marker cassette. This cassette contained the Enhanced Green Fluorescent Protein (eGFP) gene driven by the chicken β-actin promoter, and a puromycin resistance gene driven by the CAG promoter. Additionally, the selectable marker cassette was flanked by LOXP sides at its 5′ and 3′ end (Supplementary Fig. S1A). Co-Transfection of px330-sgRNA (10 μg) and AR targeting vectors (10 μg) into primordial germ cells (PGCs) was performed following previously described methods[27]. The homology regions were amplified using LSL genomic DNA (gDNA) using the following primer pairs: for the 5′homologous region, forward primer 5′- GTTGCACAGTCTCCCTGTT TTTG-3′ and reverse primer 5′- GCTCTCTTCT CTCTGCAG − 3′, for the 3′homologous region, forward primer 5′ – GTAACGCACAC AGAGAGG − 3′ and reverse primer 5′- GTCATTAGCTTAGGAAAGGT − 3′. Transfected PGCs were selected using puromycin (0.5 μg/μl) for five days to establish clonal AR[−/−] PGC populations. Before injection, AR[−/−] PGCs were resuspended in CO₂ independent Medium (ThermoFisher Scientific, USA) supplemented with 10% FBS and 1% Glutamax (ThermoFisher Scientific, USA) at a concentration of 3000 cells/μl. These cells were then injected into recipient embryos (HH 13−15) and incubated until hatching using surrogate turkey eggshell[25,26]. To eliminate the selectable marker cassette, Cre-recombination was applied directly to AR[−/−] PGCs. Highly GFP negative PGCs were sorted with a FACS sorter and implanted into recipient embryos as described above.

### PGC derivation and culturing

PGCs were collected from chicken embryonic blood at stage HH 13−15, following an established protocol[26]. The collected PGCs were cultured under controlled conditions at 37 °C and in 5% CO₂ using a defined PGC medium, as described[72].

### Genotyping of the progeny

Genomic DNA for genotyping was isolated using the ReliaPrep™ Blood gDNA Miniprep System (Promega, USA). To confirm the correct integration of the selectable marker cassette into the AR gene, genotyping was performed using primers positioned upstream of the homologous regions. A MultiPlex PCR reaction was set up with the following primers: 5′- AGTGACAACGTCGAGCACAGCT − 3′ (forward), 5′-GCTCTC CAGTTGCTCATCCTG AC − 3′, and reverse primer 5′- GTCAGACTGC TCTGCTGGAG − 3′, with an annealing temperature of 64 °C. PCR amplification was performed with 5x FIREPol® MultiPlex Mix (Solis BioDyne, Estonia) according to manufacturer's protocol.

### PCR detection of AR mRNA and mRNA of germ cell marker molecules

At embryonic day 18, gonads of AR[+/+] and AR[−/−] males and females were dissected, snap-frozen in liquid nitrogen, and stored at −80 °C. RNA was isolated from the gonads with a 'ReliaPrep™ RNA Tissue Miniprep system' (Promega) according to manufacturer's protocol. cDNA was synthesized with a 'GoScript™ Reverse Transcription Mix, Random Primer Protocol' (Promega) according to manufacturer's protocol. PCRs were performed with 'FIREPol® Master Mix with 7.5 m mM MgCl₂, 5x' according to manufacturer's protocol and with the following

primers (5′ → 3′) and PCR settings: VASA: GCTCGATATGGGTTTTGGAT (forward), TTCTCTTGGGTTCCATTCTGC (revers), 57 °C, and 40 cycles; DAZL: GCTTGCATGCTTTTCCT GCT (forward), TGCGTCA CAAAGTTAGGCA (Rev), 59 °C, and 40 cycles; AR: GAGCTGCAAA GTGTTCTTCAAGC (forward in exon 2), CAGACTGCCCAGCTTCTT CAGCTTG (revers in exon 3), 59 °C, and 35 cycles; β-actin: TACCA CAATGTACCCTGGC (forward), CTCGTCTTGTTTTATGCGC (revers), 56 °C, and 30 cycles. Each PCR contained a water control. PCR samples were loaded on a 1.5% TBE-gel with a '1 kb Plus DNA ladder' (New England Biolabs). All samples showed β-actin expression (positive control) and confirmed successful RNA isolation, cDNA synthesis, and PCRs. The observed bands were as expected, i.e. 750 bp for VASA, 536 bp for DAZL, 181 bp for AR, and 300 bp for β-actin.

### Sample preparation for LC-MS/MS

Samples were dissected from the testes of three $AR^{+/+}$ and three $AR^{-/-}$ roosters. Tissue pieces were lysed in 500 μl of Lysis-buffer (1% sodium deoxycholate, 40 mM 2-chloroacetamide (Sigma-Aldrich), 10 mM Tris(2-carboxyethyl) phosphine (TCEP; PierceTM, Thermo Fisher Scientific) in 100 mM Tris, pH 8.0) by incubation at 95 °C for 5 min and subsequent sonication using a Sonoplus sonication system (Bandelin). Samples were incubated once more at 95 °C for 5 min and once more sonicated using a Sonoplus system. Before digestion, the samples were diluted 1:1 with MS grade water (VWR). Samples were digested for 4 hrs at 37 °C with 2 μg of LysC and overnight at 37 °C with 5 μg trypsin (Promega). The solution of peptides was then acidified with Trifluoroacetic acid (Merck) to a final concentration of 1%, followed by desalting via Sep-Pak vac 3cc. Samples were vacuum dried and re-suspended in 50 μl of buffer A (0.1% Formic acid (Roth) in MS grade water (VWR)). Same amounts of the samples were then further fractionated using SCX-stage tips into six fractions. The fractions were vacuum-dried and equal amounts (approximately 200 ng) were loaded on Evotips Pure (Evosep).

### LC-MS/MS data acquisition and analysis

Evotips were eluted onto a 15-cm column (PepSep C18 15 cm × 15 cm, 1.5 μm, Bruker Daltonics) via the Evosep One HPLC system (Evosep). The column was heated to 50 °C and peptides were separated using the 30 SPD method. Using the nanoelectrospray interface, eluting peptides were directly sprayed onto the timsTOF Pro mass spectrometer (Bruker Daltonics). Data acquisition on the timsTOF Pro was performed using timsControl. The mass spectrometer was operated in data-independent (DIA) PASEF mode. Analysis was performed in a mass scan range from 100 to 1700 m/z and an ion mobility range from $1/K0 = 0.70$ Vs cm$^{-2}$ to 1.30 Vs cm$^{-2}$ using equal ion accumulation and ramp time in the dual TIMS analyzer of 100 ms each at a spectra rate of 9.52 Hz. Dia-PASEF scans were acquired a mass scan range from 350.2 to 1199.9 Da and an ion mobility range from $1/K0 = 0.70$ Vs cm$^{-2}$ to 1.30 Vs cm$^{-2}$. Collision energy was ramped linearly as a function of the mobility from 45 eV at $1/K0 = 1.30$ Vs cm$^{-2}$ to 27 eV at $1/K0 = 0.85$ Vs cm$^{-2}$. In complete, 42 diaPASEF windows were distributed to one TIMS scan each at switching Th precursor isolation windows which led to an estimated cycle time of 2.21 s. The ion mobility dimension was calibrated linearly using three ions from the Agilent ESI LC/MS tuning mix (m/z, 1/K0: 622.0289, 0.9848 Vs cm$^{-2}$; 922.0097, 1.1895 Vs cm$^{-2}$; 1221.9906, 1.3820 Vs cm$^{-2}$).

Raw data were processed using the Spectronaut 17.0 in directDIA+ (library-free) mode. Shortly, the peak list was searched against a predicted library of *Gallus gallus* from uniport (downloaded in 2023). Cysteine carbamidomethylation was set as static modification, and methionine oxidation and N-terminal acetylation as variable modifications. The match-between-run option was enabled, and peptides/proteins were quantified across samples using the label-free quantification (MaxLFQ) at the MS2 level.

### Testosterone and estrogen plasma levels

Blood samples were collected from the wing vein of the chickens, at specific ages: 1, 3, 7, 11, 15, and 20 weeks, using heparin-coated microhematocrit capillaries. The collected blood was processed to separate plasma by centrifugation at 2500 rpm for 10 min, and the plasma samples were stored at −80 °C until further analysis. Testosterone and estradiol were extracted from plasma using a modified version[73] of the partial purification on diatomaceous earth/glycol column method described[74]. In brief, the hormones were first extracted with dichloromethane and then separated based on their polarity using column chromatography. The recovery rates for testosterone and estradiol were 0.83% ± 0.05% (mean ± sd) and 0.76% ± 0.1%, respectively. Custom-made radioimmunoassays (RIA) were used for hormone measurement, following the procedures previously published[73,74]. The lower detection limit of the standard curves was determined as the first value outside the 95% confidence intervals for the zero standard ($B_{max}$). The detection limit for testosterone ranged from 0.33 to 0.39 pg/ml, while for estradiol, it was 0.23 pg/ml. All samples fell within the detectable range for testosterone. However, for estradiol, some samples were below the detection limit, and for these samples, the lowest detectable level was conservatively assigned considering the plasma volume and individual recovery value. The intra-extraction variation, determined from four assays per hormone, was 7.7% ± 3.0% for testosterone and 6.0% ± 2.4% for estradiol. The inter-assay variance was 7.5% ± 3.5 for testosterone and 13.9% ± 6.6 for estradiol.

### FSH and LH plasma levels

Serum follicle stimulating hormone (FSH) and luteinizing hormone (LH) levels were measured using competitive ELISA-based kits (Antibodies.com: chicken FSH #A74795, chicken LH #A75578). Serum samples collected at 20 weeks of age from male and female $AR^{+/+}$ and $AR^{-/-}$ were analyzed for both hormones according to the protocol provided by the company. The samples were diluted at a ratio of 1:10. The diluted samples, controls and standards were plated in duplicate on 96-well plates coated with the respective hormone. During incubation with the respective biotinylated antibody, the hormone present in the sample competed with the pre-coated antigen for the binding sites on the antibody. After incubation, the unbound antibodies were washed and the samples were incubated again with an HRP-streptavidin conjugate. After subsequent washing, TMB substrate was added and incubated again to visualize the enzymatic HRP reaction, resulting in a blue colored solution that immediately turns yellow upon addition of the acidic stop solution. The intensity of the yellow color is inversely proportional to the amount of hormone in each sample, which was then measured using a microplate reader at an absorbance of 450 nm (counting time: 1 s, delay time: 1 s). The hormone concentrations in ng/ml were then calculated using the standard curve generated by a four-parameter logistic curve (4-PL) fitted with GainData® software.

### Histology and RNAscope in-situ hybridization

Testes, ovaries and bursae of Fabricius were cryo-sectioned using a Leica CM3050 S cryostat to obtain sections of 20 μm thickness. Ten series of parallel sections were mounted on Superfrost5®Plus RNase-free slides with adjacent sections on different slides. One series for each animal and tissue was stained with a Haematoxylin or Haematoxylin-Eosin protocol (Carl Roth®, Germany). The remaining series of sections were stored at −80 °C for further analysis. In addition, for the testes and ovaries 10 μm sections of paraffin embedded material were produced for Haematoxylin-Eosin staining (Carl Roth®, Germany). Tissue samples were dehydrated in a graded series of ethanol solutions (70%, 80%, 95%, and 100%). Following dehydration, samples were cleared in xylene (Merck, 108298) and infiltrated with molten paraffin wax (Roth, Paraplast®, X880.1). Paraffin blocks were

then sectioned into slices of 10–20 μm thickness using a rotary microtome (Leica Supercut 2065). Subsequently, sections were floated on a warm water bath (48°) and collected onto glass slides (Roth, Adhesion Slides Superfrost® Plus, H867.1). Stained sections were observed and imaged using a Leica DM6000 B microscope. For measurement of diameter of the seminiferous tubules we analyzed all tubules (a total of 88 for AR$^{-/-}$, 74 for AR$^{+/+}$) of three sections per animal ($N = 3$ AR$^{-/-}$, $N = 3$ AR$^{+/+}$) using ImageJ.

For the bursae, one series from each embryo was used for immunocytochemistry with B-lymphocyte antibody Bu1(a + b) (Biozol Diagnosticsa, Eching, Germany). The sections were fixed in ice-cold acetone for 2 min, air-dried and rehydrated in PBS. After treatment with 0.3 % H$_2$O$_2$ for 30 min, the slides were washed in PBS (3 ×5 min) and then incubated in 0.025 % horse serum in PBS for 1 h at room temperature. The sections were then incubated with the primary antibody at 4 °C overnight, washed in PBS and incubated with the Vectastain® ABC kit according to the manufacturer's protocol. Peroxide detection was performed with the Vector®DAB kit. Counterstaining was performed with Haematoxylin for one minute, followed by washing in distilled water for 5 min and dehydration for coverslipping with Eukitt® mounting medium.

Of the 20-week-old animals, for one testis from each individual ($N = 3$ AR$^{++}$ and $N = 3$ AR$^{-/-}$), three sections were used for visualization of each AR mRNA, LHR mRNA (Luteinizing hormone/choriogonadotropin receptor), and DMRT1 mRNA (Doublesex and mab-3 related transcription factor 1). The sections were fixed in 4% paraformaldehyde for 1 h at 4 °C. In-situ hybridizations were performed using RNAscope® 2.5 HD Assay - Brown Kit (Advanced Cell Diagnostics, Newark, CA, USA), following the manufacturer's protocol. A chicken AR-specific probe called Gg_AR_C1 (Cat. No. 1045871-C1), was designed to detect the AR mRNA (accession number NM_001040090) in the region spanning nucleotides 1412-2373, targeting exons 3–6. A zebra finch DMRT1-specific probe called Tgu_DMRT1 (Cat. No. 1048191-C1) and a zebra finch probe called Tgu_LHCGR (Cat No. 522711-C1) were used for DMRT1 and LHR, respectively. These sections were counterstained with Haematoxylin. For AR mRNA detection in ovaries, the procedure was as above but with FITC as the fluorophore. These sections were counterstained with DAPI. For quality control, positive and negative control probes, PPIB (Cat. No. 460351) and DAPB (Cat. No. 310043) respectively, were included in the assay (Advanced Cell Diagnostics).

Stained sections were observed and imaged using a Leica DM6000 B microscope. We used ImageJ to evaluate the density of LHR mRNA positive cells among all interstitial cells of three 0.01 mm$^2$ areas per animal. For the density of LHR mRNA per cell, we counted the number of labeled mRNAs (brownish dots in Supplementary Fig. S4) of 20 cells per section of three sections per animal using ImageJ.

## Testosterone treatment of embryos

Fertilized eggs were incubated in a HEKA Favorit Olymp incubator under standard conditions, maintaining a temperature of 37.8 °C and humidity at 55%. On the third day of incubation, the eggs were partially dipped, about 3.5 cm from the pointy end, into an ice-cold 1% (w/v) testosterone-ethanol solution (Fisher Scientific GmbH, Schwerte, Germany), following a previously method described[75]. Control eggs were dipped in ice-cold ethanol-solution. After the treatment, the eggs were returned to the incubator and incubated until embryonic day 18 when embryos were euthanized and bursae were dissected. The dissected organs were immediately frozen in liquid nitrogen and stored at −80 °C until cryo-sectioning.

## Testosterone implantation and sound recordings

At one week of age, chicks were subcutaneously implanted with testosterone over the shoulder. The implants were made using 10 mm long silastic tubing from Dow Corning (USA), which was filled with crystalline testosterone (Sigma Aldrich, Germany). The tubing was sealed on both sides with silicone[76]. One day after the implantation, the chicks were equipped with lightweight wireless backpack microphones, enabling continuous individual recording of their vocalizations as separate audio channels[77]. Video cameras were also used to capture the behavior of the chicks. The recording period spanned from the day of the backpack installation until the following two weeks. At the end of the recording period, the implants were removed. A four-hour period of sound recordings were each analyzed for post-hatching days 11, 16, 18 and 20 using the software Audacity.

## Statistical procedures

The testicle weight per body weight, the diameter of Leydig cells, and the density of LHR mRNA dots per Leydig cell of adult males, the bursa weight per body weight, the comb weight, and FSH and LH plasma levels of adult males and female were compared between male genotypes and between female genotypes, respectively, with one-tailed or two-tailed t-tests. For comparison of bursa weight per body weight between genotypes of day 18 embryos, one-way ANOVA was performed. For the comparisons of the log-transformed hormone data between genotypes per age group, the fit model procedure was applied to male testosterone, female testosterone, male estradiol and female estradiol, with Standard Least Squares and incorporated REML estimation followed by LS Means Tukey HSD post hoc tests for pairwise comparisons.

## Reporting summary

Further information on research design is available in the Nature Portfolio Reporting Summary linked to this article.

## Data availability

Source data used in graphs and in the text are available in Supplementary Tables S1–S7 of the Supplementary Information and Supplementary Tables S2–S7 in Excel Format in the Source Data file. Supplementary Table S1 shows transmission rate of the knockout in the chimeras. Supplementary Table S2 shows the Mass spectrometry (LC-MS/MS) analysis of peptides of the AR of the testes of three adult homozygous (AR−/−) and three adult wild-type (AR+/+) chicken. Supplementary Table S3 shows the plasma levels of testosterone and 17ß-estradiol during ontogeny of homozygous (AR−/−), heterozygous (AR +/) and wild-type (AR+/+) male (M) and female (F) chickens. Supplementary Table S4 summarizes the of statistical analysis of testosterone and estradiol levels during development of male and female chickens. Supplementary Table S5A shows the weight of the bursa of Fabricius, of the testicles, of the comb, the body weight, the bursa-weight to body-weight ratio, the testicle-weight to body-weight ratio, and the comb-weight to body-weight ratio for homozygous (AR−/−) and wild-type (AR+/+) male and/or female chickens at adulthood. Supplementary Table S5B gives the eggs laid per week of homozygous (AR−/−), heterozygous (AR+/−) and wild-type (AR+/+) female chickens. In Supplementary Table S6, the weight of the body and bursa of Fabricius of homozygous (AR−/−), heterozygous (AR+/−) and wild-type (AR+/+) 18 days old embryos treated with testosterone or untreated are listed. Further, the bursa to body weight ratio is given. Supplementary Table S7 comprises the FSH and LH plasma levels of homozygous knockout (AR−/−) and wild-type (AR+/+) male (Supplementary Table S7A) and female (Supplementary Table S7B) chickens at 20 weeks of age. The AR$^{-/-}$ chicken strain is available upon request. Source data are provided with this paper.

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

## Acknowledgements

We thank. Mrs. M. Trappschuh and Dr. W. Goymann for conducting the radioimmunoassays, Mrs. A. Bakker and Dr. C. Voigt for performing the RNAscope labeling, Mrs. B. Steigenberger for carrying out the LC-MS/MS ysis, Mrs. M. Braun for the paraffin histology, and Dr. S. Leitner and the Animal house crew of Seewiesen to take care of the chickens.

## Author contributions

M.G. and B.S. conceived the project, K.L., M.R., A.L. C.F.-V., F.D and M.G. contributed data and analyses, T.V.L.B., A.L., D.D., R.K., S.S., H.S., L.T., and H.V. provided technical support, and M.G., B.S., A.L., K.L. M.R. and T.V.L.B. wrote the manuscript.

## Funding

## Competing interests

The authors declare no competing interests.
