## [Peer Review File · Nature Communications]

Unveiling the critical role of androgen signaling in avian sexual development.REVIEWER COMMENTS

Reviewer #1 (Remarks to the Author):

NCOMMS-23-29033

Unveiling the critical role of androgen signaling in avian sexual development.

This very interesting manuscript uses genome editing to uncover the role of androgen signaling in chicken sexual development. It provides new information to the field and exemplifies to use of CRISPR/Cas9 in the chicken, an approach which has challenges in the avian model. The authors generated Androgen Receptor knock out chickens (AR^{-/-}) and found that this caused infertility in both sexes. Some sexual dimorphisms persisted (e.g., spurs and tail feathers), other were underdeveloped in both sexes (comb, wattle, sexual behaviour). Importantly, testosterone treatment did not rescue these phenotypes, pointing to the success of the AR knockout. The data has implications for our understanding of avian sexual development, and also the recently argued hypothesis that sex development is largely cell autonomous in the chicken. The authors report “the absence of functional ARs results in a partial demasculinization of roosters and a partial defeminization of hens.” The data reveals clear roles for AR in both sexes.

Points that should be addressed are as follows.

1. Can the authors please show histology of the gonads and control birds – at embryonic stages? Also, importantly, can the authors show loss of AR protein in the gonads of the AR KO (Knockout) birds? I believe an antibody exists that recognises chicken AR (e.g, Katoh et al. (2006) Cloning and expression analysis of androgen receptor gene in chicken embryogenesis.) Also, Suppl S3 shows , quote “AR mRNA labeled by immunostainings.” Is this in situ hybridisation for mRNA or immunostaining for protein?

I think it would be very useful to show the gonadal histology in control and AR KO birds- at an embryonic time point and also post-hatching.

Previous studies from the reference above (Katoh et al.) show that AR is expressed in the embryonic gonads of both sexes, and may have a role in the ovarian cortex.

2. Can the authors show genotyping PCR to demonstrate successful disruption of AR in the birds? It is stated but not shown?

3. It would be instructive to show the expression (or lack of expression) of germ cell markers in the gonads (e.g, VASA, Gamma-H2AX, etc.).

4. The data indicate, for example, that the comb and wattle or both sexes requires AR. This is at odds with the DMRT1 knockout chickens, where ZZ birds have ovaries but still have male-like comb and wattle (unless the ovaries of ZZ-DMRT1 knockout birds make male levels of testosterone?)

5. Did the AR KO hens lay eggs?

6. It seems unlikely that non-AR dependent sexual dimorphism are largely driven by estrogen, as the authors note. In the discussion section, the authors could give further consideration to their findings in light of the concept of cell autonomous sex identity (CASI) and the curious gynandromorphic chickens, which are male on one half and female on the other half of the body. In gynandromorphs, the wattle, for example, is large on the male side

and smaller on the female side. How is this reconciled with the AR KO phenotype? Is AR required to form a wattle, then its sexual development is cell autonomous?

Reviewer #2 (Remarks to the Author):

Major comments:

This timely study presents an avian model for the study of sexual differentiation and development, specifically the role of androgens and the androgen receptor (AR). AR knockout chicken were developed using CRISPR/Cas9 technology to study the role of androgen signaling on gonad development and fertility, testosterone and estrogen production, and hormone-dependent secondary sexual structures including behavior. Results show that AR^{-/-} males and females 1) develop respective gonads and produce testosterone and estradiol similar to wildtype AR^{+/+}, but 2) are infertile; 3) have underdeveloped sexual attributes such as comb, wattles, and sexual behavior such as crowing; but some somatic traits such as body size persisted; 4) testosterone treatment of males failed to induce classical male-specific, testosterone-dependent crowing behavior and comb development, or regression of the bursa of Fabricius. The authors conclude: In birds, AR signaling is critical in fertility (germ cell development) and expression of sex-specific traits in both sexes; disruption of RA-signaling demasculinizes males and defeminizes females. These results challenge the concept of a default sex in birds and a major role of estrogens.

All experiments are well conceived and done, using proper methods, tools, and analyses.

The authors present and discuss their results in the context of previous avian studies and in relation to mammalian sex differentiation. They emphasize that their results challenge research using pharmacological doses of estrogens that prompted the “estrogen” hypothesis of avian sexual differentiation. These are very important results that shed new light onto avian sexual differentiation and the role of hormone-mediated and cell-autonomous processes. They are of broad interest across biological disciplines and fields. The study is strong, the presentation clear. I have, however, a few comments and suggestions for improvement.

Major comments:

Results/Discussion paragraph beginning line 142:

The overall pattern of testosterone secretion from hatching to sexual maturity at age 20 weeks was similar in all male and female genotypes; with an increase in males and females at age 15 – 20 weeks, when puberty occurs. The authors correctly state that this indicates presence of a functional hypothalamus-pituitary-gonadal axis in AR^{-/-} genotypes. Interestingly, AR^{-/-} males (and females) had testosterone plasma levels similar to wild-types AR^{+/+} at the time of pubertal increases. This is an important finding that deserves more emphasis. It suggests that the functional HPG axis for testosterone and its regulation by negative feedback does not involve the AR. If so, one would expect higher testosterone levels in AR^{-/-} because of a lack of feedback. Presumably, estrogen and the ER after conversion of testosterone has that function. It might be worthwhile checking the avian literature for effects of pharmacological treatment with anti-androgen (AR receptor antagonists) and anti-estrogen (aromatase inhibitor, ER receptor antagonist) for effects on

plasma testosterone levels.

This consideration is also important for the discussion of estrogenic effects in avian sexual differentiation starting on line 193.

Paragraph starting on line 169:

Please add a sentence at the end of this paragraph about the significance of the finding of absence of effect of embryo testosterone treatment on bursa Fabricius regression. Why was this experiment done? To validate that AR knockout worked?

Line 196: "estrogenic mechanisms alone are insufficient for the emergence of most sexual phenotypes..." See above comment on HPG feedback regulation.

Line 201 ff: In the discussion of the "estrogen" hypothesis (based on pharmacological doses), please consider that administration of estrogens might have interfered with feedback regulation of testosterone production (see above). You already refer to this possibility on line 206: "... the possibility of gonadal regression and reduced testosterone production." Explicitly stating here again the finding that AR $-/-$ genotypes did not have elevated testosterone levels (expected if AR would mediate feedback) could further strengthen this already strong paragraph.

Line 233: The sentence starting here is difficult to understand. Why should a plausible explanation for both, the AR-driven effects observed in this study and brain-autonomous and cell autonomous sexual development described in previous research, be "that the latter determines the hypothalamic regulatory properties that control gonadal hormone secretion."

Minor comments/suggested changes:

Line 39: challenges

Line 62: changes to androgen signaling system...

Line 65 change to ...effects of steroid hormones in birds...

Line 69: change to ... due to lack of specific experimental avian models....

Line 107: change to ... remained sexually dimorphic, with larger males, and identical weight of AR $-/-$ and AR $+/+$ roosters....

Line 113: change to ... Gonad determination ...

Line 119: change to ... lumen...

Line 127: change to ... and they did not lay eggs.

Line 137: change to ... spermatogenesis in male and

Line 188: check sentence ... for the purpose of developing crowing...?

Line 221: What do you mean with "genomic and genetic mutations?"

Line 224: change to ... ARs resulted in....

Line 232: unclear sentence: ...in male and female birds, overriding in cell-autonomous sex identity inherent to avian biology.

Congratulations to a great study, Hubert Schwabl

Dear Reviewers.

We express our sincere gratitude to you for your positive statements and insightful comments, which significantly enhanced the merit of our publication.

In the following, we address the comments point by point, implementing corresponding modifications. Please note that the revised sections in the manuscript are highlighted in yellow, both within the paper and in the accompanying revision letter.

Due to additional analysis concerning the presence of androgen receptor protein, we included two further authors.

With kind regards and on behalf of all authors,

Manfred Gahr

Reviewer #1

1A. Can the authors please show histology of the gonads and control birds – at embryonic stages? I think it would be very useful to show the gonadal histology in control and AR KO birds- at an embryonic time point and also post-hatching. Previous studies from the reference above (Kato et al.) show that AR is expressed in the embryonic gonads of both sexes, and may have a role in the ovarian cortex.

Supplement, lines 73-88. New Fig. S4. A functional androgen receptor is not required for testicular and ovarian development in male and female chicken embryos on day 18. Histological analyses, along with the examination of DMRT1 gene expression, a marker for Sertoli cells, reveal comparable features in the testis of a wild-type ($AR^{+/+}$; A), a heterozygous ($AR^{+/-}$; B) and a homozygous ($AR^{-/-}$; C) male. DMRT1 mRNA expressing cells surround the developing seminiferous tubules (ST) in each genotype (A-C). The seminiferous tubules were distributed throughout the testes and some had already a lumen. Similarly, histological examination shows uniform ovarian organization and distribution of the gene aromatase, a marker for granulosa cells, in the left ovary of a wild-type ($AR^{+/+}$; D), a heterozygous ($AR^{+/-}$; E) and a homozygous ($AR^{-/-}$; F) female embryo. Pre-granulosa cells expressing aromatase mRNA are localized in the zone of the medulla (M) adjacent to the ovarian cortex (C). The brownish dots represent mRNA labeling for aromatase mRNA (D-F) and DMRT1 (A-C), overlaying the presumed pre-granulosa cells and Sertoli cells, respectively. The mRNA detection was performed using the RNAscope method and counterstained with Hematoxylin (bluish). (*) in A indicates a staining artefact. "L" indicates the lacunar channels. Coronal sections with ventral orientation at the top in relation to the anterior-posterior body axis. Scale bar represents 100 μ m.

Additional Figure (not currently included in the supplement of the revision): A functional androgen receptor is not required for testicular and ovarian development at the third post-

hatching week. Histology and expression of the gene DMRT1, a marker for Sertoli cells, in the testis of a heterozygous ($AR^{+/-}$; **A**) and a homozygous ($AR^{-/-}$; **B**) male were comparable. DMRT1 mRNA expressing cells, likely Sertoli cells, surround the developing seminiferous tubules (ST) in both genotypes (A, B). The seminiferous tubules were distributed throughout the testes and all had a lumen. Lacunar channels were much reduced at this age compared to the 18 days embryonic age (Fig. S4). Accordingly, in females the left ovary of a wild-type ($AR^{+/+}$; **C**) and a homozygous ($AR^{-/-}$; **D**) showed comparable histology and distribution of the gene aromatase, a marker for granulosa cells. The presumable pre-granulosa cells labeled for aromatase mRNA are still partially located in the dorsal zone of the medulla (M), adjacent to the ovarian cortex (C). Due to the folding of the ovary (somewhat more advanced in the $AR^{-/-}$ female), the pre-granulosa cell layer appears between the cortical layers. Further, pre-granulosa cells seem to migrate into the cortex towards the follicles, resulting in a pre-granulosa layer partially surrounding some follicles (Fo*), while others (Fo) remain devoid of pre-granulosa cells. Lacunar channels are notably reduced compared to the 18-day embryos (Fig. S4). The labeling of aromatase mRNA (C, D) and DMRT1 mRNA (A, B) appear as brownish dots overlaying the presumed pre-granulosa cells and Sertoli cells, respectively. The mRNA detection was performed using the RNAscope method and counterstained with Hematoxylin (bluish). Coronal sections with ventral orientation at the top in relation to the anterior-posterior body axis. The scale is 100 μ m.

Since the figures suggest that the ontogenetic development of the ovary and the testicles in homozygous AR knockouts and wild-type chickens appears largely normal until three weeks post-hatching, we insert the following into the paper at lines 127-132: This deficient development (Fig. 1A4) becomes apparent later in ontogeny, suggesting a dependence on androgen receptor (AR)-sensitive mechanisms that operate during puberty, as testicular development of $AR^{-/-}$ and $AR^{+/+}$ males showed comparable progression at embryonic age (Fig. S4A-S4C) and at three weeks after hatching (unpublished data), despite the presence of AR expression in the testes during embryonic age (Kato et al., 2006). Insert into the paper at lines 142-147: While AR is expressed in the ovary during embryonic age (Kato et al., 2006), our data suggests that androgen signaling may not play a crucial role in the organization of the ovary until puberty. Despite the sensitivity of aromatase expression to AR action in the ovary (Kato et al., 2006), systemic inactivation of this signaling pathway in our knockout chickens does not down-regulate aromatase (Fig. S4D-S4F) so that sex chords develop in the $AR^{-/-}$ females.

We suggest to include only the new Fig. S4 into the current publication to avoid overloading it with gonadal development details. Instead, we intend publishing this additional data separately at a later time point. However, we are open to including it if the reviewers find it beneficial for a better comprehensive understanding.

1B. Also, importantly, can the authors show loss of AR protein in the gonads of the AR KO (Knockout) birds? I believe an antibody exists that recognises chicken AR (e.g, Katoh et al. (2006) Cloning and expression analysis of androgen receptor gene in chicken embryogenesis.)

AR Antibodies are all based on mammalian sequences, frequently of exon 1, which is little conserved in evolution. In relation, all antibodies that we tested so far showed little specificity for avian ARs. Further, the specific antibody mentioned in Katoh et al, 2006 is unfortunately no longer available. For these reasons, we opted for Mass-Spectrometry (LC-MS/MS) analysis to verify if a truncated AR protein exists. This approach, performed by the Mass-Spectrometry facility at the Max Planck Institute for Biochemistry, is detailed now in the methods section, included at lines 545-582 (Methods: **Sample preparation for LC-MS/MS and LC-MS/MS data acquisition and analysis**).

The Mass-Spectrometry analysis involved the examination of gonads from three adult male AR^{-/-} and three adult AR^{+/+} chicken. With this procedure we discovered the presence of peptides in exon 1 for each individual. However, peptides related to exon 2 to exon 8 were not detected in the knockouts. This strongly confirms the lack of a functional AR in the AR^{-/-}, confirming a protein truncation following the knockout of exon 2. This finding aligns with the chicken AR sequence, where the removal of exon 2 leads to a stop codon at 966 nucleotides, resulting in a truncated protein of 322 amino acids. This truncated AR lacks the DNA binding domain (exon 2) and the steroid-binding domain (exon3). The results of LC-MS/MS are summarized in a new Tab. S2:

Peptide Sequence	AR Exons	genotypes					
		AR+/+	AR+/+	AR+/+	AR-/-	AR-/-	AR-/-
APRPDAAEPPEPPAPAAFK	1		273.2	260.6	326.9	355.1	283.6
DCFVLPPPAR	1			35.7		52.8	50.4
EPPPREDCMFALPGGPPR	1	136.4	47.0	34.3			
GSGAEAAALAVEVPAGLPLYR	1	55.9		35.3		139.1	93.5
VSPPEEPPGR	1					272.9	
SELGPAWEGYAGAYGDVR	1	157.2	64.8	66.4		31.2	
EHILPIDYYFPPQK	2	149.2	77.1	79.7			
TCLICGDEASGCHYGALTCGSK	2	8.1	28.3	3.8			
CYEAGMTLGAR	3 & 4	247.6	140.2	89.1			
MLYFAPDLVFNEYR	5 & 6	564.3	176.8	102.7			
VLDSVHPIAK	7 & 8	31.5	26.3	20.8			
DLHQFTFDLLIK	8	94.1	45.5	39.1			
AHMVSVDYPEMMAEIIISVQVPK	8	368.8	95.0	112.2			
VKPIYFHAE	8	193.6	139.5	49.1			

Supplement, lines 137 to 144: Tab. S2. The androgen receptor (AR) of the AR^{-/-} chicken is a truncated protein. Mass spectrometry (LC-MS/MS) analysis of peptides of the AR of the testes of three adult AR^{-/-} and three adult AR^{+/+} chicken. The detection of peptides corresponding to exon 1 in all individuals, coupled with the absence of peptides referencing to exons 2 to 8 in the knockouts, indicate the truncation of AR rendering it non-functional in this chicken strain, lacking the DNA-binding domain and the -binding domain. Listed are all peptides detected in coding exons 1 to 8 of the AR. Numbers indicated the peptide quantities (AU). Empty cells indicate the lack of detection of these peptides in a chicken.

We state now in lines 90-95: In homozygous AR^{-/-} chicken, our analysis revealed the absence of AR mRNA including exon 2 and exon 3 in embryonic gonads. (Fig. S2, PCR analysis) and the absence of peptides related to exons 2 to 8 in adult testes (Tab. S2; LC-MS/MS analysis). This confirms that the loop-out of exon 2 (Fig. S1) resulted in a truncated AR protein, which included only exon 1 and, therefore, was non-functional, lacking the DNA binding domain (exon 2) and the hormone binding domain (exon 3).

1C. Also, Suppl S3 shows, quote “AR mRNA labeled by immunostainings.” Is this in situ hybridisation for mRNA or immunostaining for protein?

Sorry for the wrong wording: The labeling is by in situ hybridization for AR mRNA. We corrected the sentence accordingly: Supplement, at line 95: (B), AR mRNA labeled by RNAscope in-situ hybridization (indicated by brown dots) was detected in the Sertoli cells.

2. Can the authors show genotyping PCR to demonstrate successful disruption of AR in the birds?
It is stated but not shown?

We refer to the new Figure S2C (see above, Query 1B and below Query 3): AR was not detected in the gonads of male and female chickens using primers directed for exon 2 (See Query 3 for PCR methods). We refer to this new figure now in lines 90-95: In homozygous AR^{-/-} chicken, our analysis revealed the absence of AR mRNA including exon 2 and exon 3 in embryonic gonads. (Fig. S2, PCR analysis) and the absence of peptides related to exons 2 to 8 in adult testes (Tab. S2; LC-MS/MS analysis). This confirms that the loop-out of exon 2 (Fig. S1) resulted in a truncated AR protein, which included only exon 1 and, therefore, was non-functional, lacking the DNA binding domain (exon 2) and the hormone binding domain (exon 3).

3. It would be instructive to show the expression (or lack of expression) of germ cell markers in the gonads (e.g, VASA, Gamma-H2AX, etc.).

Both wild-type ($AR^{+/+}$) and homozygous ($AR^{-/-}$) males and females showed expression of the germ cell markers VASA and DAZL (Tsunekawa et al., 2000; Lee et al., 2015), indicating the functional presence of primordial germ cells in the gonads, unaffected by the AR knockout. We now specify in line 132-133: Complete infertility was evident, as no mature sperm were found in $AR^{-/-}$ roosters while germ cells were present in their testes (Fig. S2A, S2B). Additionally, in lines 139-141: We note that, akin to the testes, the germ cell markers VASA and DAZL were also present in the ovaries of $AR^{-/-}$ females (Fig. S2A, S2B). These findings are now illustrated in the new Figure S2:

Supplement, at lines 46-52: new **Fig. S2:** Analysis of the expression of VASA, DAZL, AR, and β -actin mRNA in the gonad of male and female 18-day embryos, detected by PCR. Both, wild-type ($AR^{+/+}$) and homozygous knockout ($AR^{-/-}$) male and female gonads showed expression of the germ cell markers VASA and DAZL. These results indicate the presence of primordial germ cells in the testis and ovary, unaffected by the AR mutation. Water served a negative control and β -actin as a positive control. Given that the AR mutation was made by looping out exon 2, the AR mRNA

was expected being absent in AR^{-/-} samples with the used primer pair for exons 2 and 3. The PCR methods are detailed in lines 526-543.

4. The data indicate, for example, that the comb and wattle or both sexes requires AR. This is at odds with the DMRT1 knockout chickens, where ZZ birds have ovaries but still have male-like comb and wattle (unless the ovaries of ZZ-DMRT1 knockout birds make male levels of testosterone?)

Regrettably, the authors of the ZZ-DMRT1 knockout study (Ioannidis et al, 2021) did not provide information on testosterone levels in their birds, which is somewhat surprising. As discussed in the revised lines 254-277, we propose that the most straightforward interpretation of our current findings, as well as those from the ZZ-DMRT1 knockout chicken study, would be the production of male-typical levels of testosterone in the latter. Within this context, we suggest that brain-autonomous mechanisms may play a role in guiding the male-typical production of testosterone.

5. Did the AR KO hens lay eggs?

We rephrased the sentence in line 139: Among adult females, AR^{-/-} females had smaller ovaries than AR^{+/+} and AR^{+/-} females due to the absence of late-stage (so called hierarchical) follicles and ovulated oocytes (Fig. 1C3, 1C4 1D3, 1D4; Fig. S3), and they did not lay eggs (Fig. 2F).

6. It seems unlikely that non-AR dependent sexual dimorphism are largely driven by estrogen, as the authors note. In the discussion section, the authors could give further consideration to their findings in light of the concept of cell autonomous sex identity (CASI) and the curious gynandromorphic chickens, which are male on one half and female on the other half of the body. In gynandromorphs, the wattle, for example, is large on the male side and smaller on the female side. How is this reconciled with the AR KO phenotype? Is AR required to form a wattle, then its sexual development is cell autonomous?

We extended the discussion on lines 257 to 265: In relation to this, gynandromorphic birds display distinct male and female characteristics on opposite halves of their bodies [5]. Such chickens with a predominantly male-typical wattle on one side and a predominantly female-typical wattle on the other exemplify the androgenic activation of body-half-specific genetic mechanisms. Either the androgen-induced cell proliferation or the AR abundance could be different between male and female wattle tissue in gynandromorphic birds. As shown in the present study, sexual dimorphism persists in head ornamentation, such as the combs of AR^{-/-} males and females (Fig. 1; Tab. S5), supporting the assumption that the tissues respond to AR signaling in a sex-specific, cell-autonomous manner. Related to this, we included in lines 110-112

of the results: Interestingly, the comb weights of AR^{-/-} roosters and AR^{-/-} hens remained sexually dimorphic ($t(6) = 5.69$, $p = 0.002$, one-tailed t-test; Tab. S5).

Reviewer #2:

1. Results/Discussion paragraph beginning line 142: The overall pattern of testosterone secretion from hatching to sexual maturity at age 20 weeks was similar in all male and female genotypes; with an increase in males and females at age 15 – 20 weeks, when puberty occurs. The authors correctly state that this indicates presence of a functional hypothalamus-pituitary-gonadal axis in AR^{-/-} genotypes.

Interestingly, AR^{-/-} males (and females) had testosterone plasma levels similar to wild-types AR^{+/+} at the time of pubertal increases. This is an important finding that deserves more emphasis. It suggests that the functional HPG axis for testosterone and its regulation by negative feedback does not involve the AR. If so, one would expect higher testosterone levels in AR^{-/-} because of a lack of feedback. Presumably, estrogen and the ER after conversion of testosterone has that function. It might be worthwhile checking the avian literature for effects of pharmacological treatment with anti-androgen (AR receptor antagonists) and anti-estrogen (aromatase inhibitor, ER receptor antagonist) for effects on plasma testosterone levels. This consideration is also important for the discussion of estrogenic effects in avian sexual differentiation starting on line 193.

We agree to better emphasis that the HPG axis is functional in the knockouts. However, the underlying mechanisms need to be seen. It seems unlikely that the estrogens and the estrogen receptors are the mechanism for the feedback control of hormone production of the HPG axis since estrogen levels remain low throughout development and puberty, while testosterone levels increase. However, a plausible mechanism could involve the local aromatization of testosterone in the hypothalamus, with subsequent estrogen signaling to GnRH neurons, as suggested by the reviewer. Unfortunately, most, even recent experiments on this topic (e.g. Baghel and Srivastava, *Theriogenology*. 2020 Oct 1;155:98-113) affect testosterone production with systemic procedures involving very high levels of hormones so that low levels of testosterone following this endocrine manipulations could as well be explained by effects in the testes itself (e.g. Ref. 60, Lambeth et al., 2016) or could be due to uncontrolled site effects. Further, other hormones of the HPG axis such as LH could as well function as a feedback molecule. To keep the discussion concise, we included the following statements in lines 170-175: This functionality does not appear to be mediated by the AR or by feedback mechanisms with gonadal estrogens, as

testosterone production increases while plasma estrogen levels remain constant throughout puberty in males (Fig. 2A, 2C). However, the aromatization of testosterone in the hypothalamus and the signaling of these estrogens to GnRH neurons may be involved in the hypothalamic-pituitary-gonadal axis in the chicken.

We further state in lines 268-271: To discern the relative contributions of androgenic, estrogenic and cell-autonomous mechanisms of sexual phenotypes, an effective approach would be the local inactivation of estrogen receptors and aromatase in tissues of AR^{-/-} males and females.

2. Paragraph starting on line 169: Please add a sentence at the end of this paragraph about the significance of the finding of absence of effect of embryo testosterone treatment on bursa Fabricius regression. Why was this experiment done? To validate that AR knockout worked?

We included in lines 202-205: The results not only affirm the successful elimination of the AR but also indicate the significance of androgen-regulated immune system development in avian sexual differentiation—an aspect that has been overlooked in recent decades.

3. Line 196: “estrogenic mechanisms alone are insufficient for the emergence of most sexual phenotypes...” See above comment on HPG feedback regulation.

Indeed, may be the HPG axis is one phenotype that is under estrogenic control (see as well the discussion of Query 1). To solve this question in the future we suggest to include in lines 268-270: To discern the relative contributions of androgenic, estrogenic and cell-autonomous mechanisms of sexual phenotypes, an effective approach would be the local inactivation of estrogen receptors and aromatase in tissues of AR^{-/-} males and females.

4. Line 201 ff: In the discussion of the “estrogen” hypothesis (based on pharmacological doses), please consider that administration of estrogens might have interfered with feedback regulation of testosterone production (see above). You already refer to this possibility on line 206: “... the possibility of gonadal regression and reduced testosterone production.” Explicitly stating here again the finding that AR^{-/-} genotypes did not have elevated testosterone levels (expected if AR would mediate feedback) could further strengthen this already strong paragraph.

Thanks again for the insightful comments. We included in line 235-239: The demasculinized phenotype of estrogen-treated avian embryos could be attributed to various factors, such as pharmacological disruption of testosterone production in the gonad, downregulation of gonadotropin signaling to the gonad due to the negative feedback of estrogens in the hypothalamus, or a cell type-specific alteration of AR expression in the brain.

5. Line 233: The sentence starting here is difficult to understand. Why should a plausible explanation for both, the AR-driven effects observed in this study and brain-autonomous and cell autonomous sexual development described in previous research, be “that the latter determines the hypothalamic regulatory properties that control gonadal hormone secretion.”

We rephrased (in lines 274-277) this sentence to “the latter might determine the properties of the hypothalamus that control the sex-specific production of gonadal hormone levels”. This emphasizes that the sex-specific production of gonadal hormone profiles depends on the hypothalamus rather than the gonad itself. As mentioned earlier, this mechanism could also account for the development of male phenotypes in ZZ-DMRT1 knockouts, in addition to the responsiveness of target tissues to testosterone.

6. Minor comments/suggested changes:

We corrected all minor suggestions. In former line 221 (now line 251) we deleted “genetic and genomic”. The unclear statement in line 232 was deleted and the last paragraph rephrased, lines 254-277 (see as well Query 6 of Reviewer 1) and Query 5 of Reviewer 2.

REVIEWER COMMENTS

Reviewer #2 (Remarks to the Author):

The authors have addressed all of my concerns and comments on the previous version and made relevant changes in the current manuscript.

Reviewer #3 (Remarks to the Author):

Despite generating a knockout bird for androgen receptor, the authors do not provide a thorough characterization of the phenotypes. I suggest a stronger characterization of the phenotypes provided as the manuscript at this state is more descriptive.

Major comments:

The expansion of the interstitium in the knockout could reflect an expansion of steroidogenic Leydig cells. Do you see more Leydig cell differentiation in the knockout testis? Or are they the non-steroidogenic interstitial cells? This need characterization.

Showing RT-PCRs is not enough to evidence the germ cell phenotype. Please provide a characterization of the testicular cords showing the differences between wild type and KO spermatogenesis. When is this process arrested in the germ line? The same applies for the ovary.

Similarly, the histological sections in Figure 1 and S4 are not convincing enough of the gonadal phenotype. Please provide better high-quality images including low and high magnification. The inclusion of cell type specific markers is also required to characterize the potential effect in infertility.

Based on the methodology, AR seems not to be required for spermatogenesis. The authors mention that they generated in vitro AR^{-/-} germ cell lines and reintroduce them into surrogate hosts. If AR is required for spermatogenesis, no offspring should carry the knockout allele, as they won't generate functional sperm. Instead, using these birds they were able to generate AR ^{+/-} offspring crossing with wild-type hens (AR ^{+/+}). This suggest that the gonadal environment in the global AR ^{-/-} birds is not optimal for germ cell differentiation, rather than the germ cells being affected. The text should be modified accordingly, and these results should be discussed.

The justification on line 170 that the hypothalamus-pituitary-gonad is functional is not adequate. Please provide information regarding the levels of FSH and LH to support this statement. The expansion of the interstitium could signify Leydig cell hyperplasia, which could be due to higher levels of LH. In addition, abnormal Sertoli development/spermatogenesis, as well as lack of follicle maturation could also signify incorrect FSH signaling.

What is the reason behind androgen treatment to study the crowing behavior in the androgen receptor knockout, if the AR knockout animals are irresponsive to androgens. What happens if you don't induce the crowing with androgens? Do the knockout birds start crowing later than the ^{-/+} or ^{+/+} animals? Do they ever crow?

The same applies to the testosterone treatment in the bursa and the comb growth. It would be more significant to show the actual differences between wild type and knockout. For example, a time course of the normal bursa regression in +/+ males compared with the -/- males. Supplementing the bursa data with apoptosis markers, like cleaved caspase 3, would strengthen the phenotype, making it consistent with the idea proposed.

Minor comments:

Figure 2. Why you decided to run one-tailed t-test instead of ANOVA when more than one condition is present? Please provide a description of the analysis performed in these figures.

In figure 1, it is stated that the scale bars represent figures in E panels, but there are only A-D panels.

The Lacunar channels in figure S4 A to C seems to be a sectioning artefact rather than lacunar channels.

Discussion title is missing.

Results sub-titles are also missing.

Response to Reviewer # 2:

We thank Reviewer 2 for his final agreement.

Responses to Reviewer #3:

We would like to thank the reviewer for his comments and are confident that our additional analyses and statements have further improved the paper. The modifications are labeled in yellow throughout the paper and in the response letter.

1. Despite generating a knockout bird for androgen receptor, the authors do not provide a thorough characterization of the phenotypes. I suggest a stronger characterization of the phenotypes provided as the manuscript at this state is more descriptive: This paper focuses on investigating the role of androgen receptors (ARs) in the sexual differentiation of male and female birds. We respectfully disagree with the assertion that we did not adequately characterize the phenotypes of the knockout males and females. In our manuscript, we thoroughly explored various aspects related to sexual differentiation, including vocal behavior, external sexual ornaments, reproductive functions, Bursa development, gonadal development, gonadal endocrine system, and overall body development. Nevertheless, we added further measurements on the testicles, which were the focal point of several queries raised by Reviewer 3 (see Points 2 to 6). Since ARs are expressed in many somatic tissues, we recognize that further phenotypes than those described in our paper could be studied, indeed. However, this is beyond the scope of the paper but could be done by the interested scientific community since the AR knockout chicken are available for others upon request.

2. The expansion of the interstitium in the knockout could reflect an expansion of steroidogenic Leydig cells. Do you see more Leydig cell differentiation in the knockout testis? Or are they the non-steroidogenic interstitial cells? This need characterization. We characterized the interstitium of the testis by labeling the luteinizing hormone receptor (LHR), a marker for Leydig cells, to gain insights into their dynamics. In relation we included a new Supplementary Figure S4. The expansion observed in the interstitium is due to proliferation of both steroidogenic Leydig cells and non-steroidogenic interstitial cells which do not express the LHR. Further, the expression of LHR per Leydig cell strongly increased. Additionally, we used DMRT1 mRNA as a marker for Sertoli cells. The expression of DMRT1 mRNA was comparable in the wild-type ($AR^{+/+}$) and knockout roosters ($AR^{-/-}$) with localization restricted to the Sertoli cells and to spermatogonia type cells of the seminiferous tubules. Accordingly, we changed the text in lines 133 – 141: Specifically, we observed a significantly smaller diameter of the seminiferous tubules ($AR^{+/+}$: $452 \pm 49 \mu\text{m}$; $AR^{-/-}$: $211 \pm 55 \mu\text{m}$; $t(4) = 5.67$, $p = 0.002$, one tailed t-test, Fig. 1). The proportion of interstitial tissue was $9.8 \pm 3.7\%$ in $AR^{+/+}$ testicles and $60.7 \pm 8.9\%$ in $AR^{-/-}$ testicles (see Fig. S4). The expanded interstitial tissue is composed of Leydig cell-like cells expressing luteinizing hormone (LH) receptor (LHR) mRNA and of interstitial cells not expressing LHR (Fig. S4). The much higher expression of LHR receptors in Leydig cells of the $AR^{-/-}$ roosters as compared to $AR^{+/+}$ roosters ($AR^{+/+}$: 1.5 ± 0.16 dots per cell; $AR^{-/-}$: 4.46 ± 0.39 dots per cell; $t(4) = 12.24$, $p = 0.0001$, one tailed t-test; Fig. S4) as well as the hyperplasia of these cells suggest permanent high levels of LH, as found in mammalian species [31].

Further, since LH levels were not elevated (see query 6) we state in lines 142-144: However, plasma LH levels were similar between $AR^{-/-}$ and $AR^{+/+}$ roosters (one-sided t-test, $p > 0.05$, see Tab. S7) reminiscent of rare cases of Leydig cell hyperplasia and low LH levels in AR-deficient humans [32].

Fig. S4. Hyperplasia of interstitial tissue including Leydig cell-like cells expressing luteinizing hormone receptors (LHR) in the seminiferous tubules (ST) of homozygous $AR^{-/-}$ males. Histological analyses, along with the examination of expression of DMRT1 mRNA expression (brownish stain), a marker for Sertoli cells, reveal comparable levels of DMRT1 mRNA between wild-type ($AR^{+/+}$; A) and $AR^{-/-}$ (B) males. In contrast, the expression level of LHR mRNA, indicative of Leydig cells, markedly differs between the two genotypes, with low levels in $AR^{+/+}$ (brownish dots indicated by arrowheads in the insert c of C) and higher in $AR^{-/-}$ (arrowheads in d of D). The mean densities of dots representing LHR mRNA were 1.63, 1.55 and 1.32 for three $AR^{+/+}$ and were 4.79, 4.54 and 4.03 for three $AR^{-/-}$ males, respectively. Further, interstitial tissue is massively expanded in $AR^{-/-}$ (B, D), best visible in the upper part of D (arrows) and in d (arrows), and contains many LHR mRNA expression cells. In A to D, the small squares indicate the enlarged areas of the inserts a-d. The detection of LHR mRNA and DMRT1 mRNA was performed using the RNAscope method and counterstained with Hematoxylin (bluish). The scale bar represents 150 μm for A-D and 50 μm for a-d.

3. Showing RT-PCRs is not enough to evidence the germ cell phenotype. Please provide a characterization of the testicular cords showing the differences between wild type and KO spermatogenesis. When is this

process arrested in the germ line? The same applies for the ovary. As previously mentioned, in $AR^{-/-}$ the testicles don't form mature sperms but spermatogonia are maintained and the ovaries form small follicles but lack the hierarchical follicles. Accordingly, the maturation of germ cells seems to end before the meiosis. We included inserts into Fig. 1A4 and Fig. 1B4, showing parts of the seminiferous tubules at higher magnification, and introduced a new figure, Fig. S5, which depict the inserts of Fig. 1 in high magnification. The follicles of knockout and wildtype were already clearly visible in the previous version of Fig. 1C4 and Fig. 1D4, so we did not change these figures parts. In the $AR^{-/-}$ males, spermatogonia divide and primary spermatocyte form, but only few spermatids differentiate, as shown in the insert of Fig. 1A4. Only ca 5% of the seminiferous tubules contain rare spermatids, as shown in Fig. 1A4 and new Fig. S5A, and don't contain mature sperms (insert of Fig. 1B4 and Fig. S5B). Accordingly, we modified the text in lines 151-154: While spermatogonia and primary spermatocytes were abundant in $AR^{-/-}$ seminiferous tubule, the formation of spermatids was markedly limited, observed in minimal quantities only in about 5% of the tubules (insert in Fig. 1A4, Fig. S5A) compared to $AR^{+/+}$ tubules (insert in Fig. 1B4, Fig. S5B).

We discuss this result in lines 162-168: In particular, AR appears to be relevant for the transition from primary spermatocytes to spermatids. In mouse models with Sertoli cell-specific AR ablation, meiosis and thus spermatogenesis are completely blocked [34]. The observed residual spermatogenesis activity in $AR^{-/-}$ chickens could be due to high follicle stimulating hormone (FSH) levels, which induce some of the otherwise AR-dependent mechanisms of Sertoli cells [35]. Indeed, FSH plasma levels were significantly elevated in $AR^{-/-}$ males as compared to $AR^{+/+}$ males (one-tailed t-test, $t(8) = 2.233$, $p = 0.028$; Tab. S7).

New **Fig. S5: Comparison of seminiferous tubules in $AR^{-/-}$ and $AR^{+/+}$ testicles.** The seminiferous tubules of the $AR^{-/-}$ testicles (A) lack a lumen and spermatogenesis is interrupted compared to the $AR^{+/+}$ testicles (B). In A, only few round spermatids (RS) and elongated spermatids (ES) are observed, with an absence of mature spermatozoa (SZ) typically found in abundance in the wild-type testicle (B). Additional abbreviations: SC = Sertoli cell, SG = spermatogonium, SP = primary spermatocyte. Sections were stained with Hematoxylin. Scale bar represents 80 μ m for A and B.

4. Similarly, the histological sections in Figure 1 and S4 are not convincing enough of the gonadal phenotype. Please provide better high-quality images including low and high magnification. The inclusion of cell type specific markers is also required to characterize the potential effect in infertility. As stated above (Point 3), we improved the photomicrographs in Fig. 1A4 and 1B4 and included a new Fig. S5 to provide detailed histological views of the testicles at higher magnification. Since the different cell types within the germ cell line are clearly distinguishable histologically, as shown by the work of Estermann and colleagues (Estermann, Major & Smith, 2021, Genes 12, 1459), we have not used specific markers for this (Fig. 1A, 1B; Fig. S5), but have used such markers for Sertoli cells and Leydig cells. As previously noted, the knockout effect results in the absence of spermatogenesis, which ends mainly before or with spermatid formation in males and prior to hierarchical follicle development in females. We extended the related results and discussion, lines 151-154: **While spermatogonia and primary spermatocytes were abundant in AR^{-/-} seminiferous tubule, the formation of spermatids was markedly limited, observed in minimal quantities only in about 5% of the tubules (insert in Fig. 1A4, Fig. S5A) compared to AR^{+/+} tubules (insert in Fig. 1B4, Fig. S5B).**

5. Based on the methodology, AR seems not to be required for spermatogenesis. The authors mention that they generated in vitro AR^{-/-} germ cell lines and reintroduce them into surrogate hosts. If AR is required for spermatogenesis, no offspring should carry the knockout allele, as they won't generate functional sperm. Instead, using these birds they were able to generate AR^{+/-} offspring crossing with wild-type hens (AR^{+/+}). This suggest that the gonadal environment in the global AR^{-/-} birds is not optimal for germ cell differentiation, rather than the germ cells being affected. The text should be modified accordingly, and these results should be discussed. We thank the reviewer's observation: The primordial germ cells introduced had indeed an AR^{-/-} genotype. Consequently, we can indeed conclude that AR^{-/-} roosters lack a gonadal environment conducive to functional spermatogenesis. We have modified the text accordingly (lines 157-168): **The PGC used to generate the knockout line had an AR^{-/-} genotype and were injected into an AR^{+/+} gonadal environment of the recipient rooster, resulting in the development of functional AR^{-/-} spermatozoa. This underlines that AR expression in the germ cell line does not play a role in spermatogenesis, but that AR expression of Sertoli cells is crucial for the induction of signalling supporting spermatogenesis in the chicken. In particular, AR appears to be relevant for the transition from primary spermatocytes to spermatids. In mouse models with Sertoli cell-specific AR ablation, meiosis and thus spermatogenesis are completely blocked [34]. The observed residual spermatogenesis activity in AR^{-/-} chickens could be due to high follicle stimulating hormone (FSH) levels, which induce some of the otherwise AR-dependent mechanisms of Sertoli cells [35]. Indeed, FSH plasma levels were significantly elevated in AR^{-/-} males as compared to AR^{+/+} males (one-tailed t-test, $t(8) = 2.233$, $p = 0.028$; Tab. S7).**

6. The justification on line 170 that the hypothalamus-pituitary-gonad is functional is not adequate. Please provide information regarding the levels of FSH and LH to support this statement. The expansion of the interstitium could signify Leydig cell hyperplasia, which could be due to higher levels of LH. In addition, abnormal Sertoli development/spermatogenesis, as well as lack of follicle maturation could also signify incorrect FSH signaling. We determined the functionality of the hypothalamus-pituitary-gonad axis by analyzing the ontogeny of testosterone and estrogen profiles, which is similar in wild type and knockout (see as well the supportive comment of Reviewer 2 to this point). Nevertheless, as requested by Reviewer 3, we further analyzed the presence of steroidogenic Leydig cells in the testis and found a significant increase in Leydig cells like cells in adult testis using LHR mRNA as a marker. The density of Sertoli cells, seemed, however, not affected (see above **new Fig. S.4**)

We further measured the FSH and LH levels in AR^{-/-} and AR^{+/+} males and females using a chicken ELISA (see

methods, lines 696-713). FSH levels are significantly elevated in male and female knockout but LH levels were similar in wildtype and knockout males and females. This information is now included in

lines 142-144: However, plasma LH levels were similar between AR^{-/-} and AR^{+/+} roosters (one-tailed t-test, $p > 0.05$, see Tab. S7) reminiscent of rare cases of Leydig cell hyperplasia and low LH levels in AR-deficient humans [32].

lines 167-168: Indeed, FSH plasma levels were significantly elevated in AR^{-/-} males as compared to AR^{+/+} males (one-tailed t-test, $t(8) = 2.233$, $p = 0.028$; Tab. S7).

lines 187-189: In relation to the disrupted fertility, FSH plasma levels were significantly elevated in AR^{-/-} females as compared to AR^{+/+} females (one-tailed t-test, $t(8) = -6.441$, $p = 0.0001$; Tab. S7).

We state further in lines 192-197: An AR-sensitive signalling mechanism of gametogenesis concerns the components of the activin/inhibin signalling pathway that regulates FSH production in the pituitary gland [43], which could lead to elevated FSH levels in the AR^{-/-} chicken. Since testosterone levels are similar in AR^{-/-} and wildtype chickens (see next paragraph), the lack of a functional AR in gametogenesis cannot be compensated for by AR independent testosterone driven or non-steroidal mechanisms.

From these new data we suggest now the following scenario for steroid production in knockout animals, lines 215-237: If we assume that chicken Leydig cells' steroidogenesis is LH and AR-regulated as in mammals [44], we expect the functioning hypothalamic-pituitary-gonadal axis should lead to high LH release due to initially low testosterone production [31]. However, LH (Tab. S7) and testosterone plasma levels (Fig. 2A) were similar between AR^{-/-} and AR^{+/+} males, while we observed high expression of LHR mRNA in Leydig cells and substantial proliferation of LHR mRNA-expressing cells in the testicular interstitium (Fig. S4). Thus, it appears that each LHR-positive cell of AR^{-/-} birds constitutively produces low testosterone levels, leading to normal elevated testosterone levels in adult AR^{-/-} males due to the sheer number of these cells. Thus, the hypothalamic-pituitary-gonadal axis of AR^{-/-} males actually functions through a different mechanism (hyperplasia of steroidogenic testicular cells) compared to AR^{+/+} roosters (regulation of LH production). In AR mutant humans, too, testosterone production is rather normal [31]. What then triggers the proliferation of LHR expressing cells in AR^{-/-} roosters needs to be seen. Future, detailed analysis of the expression of all major components of the hypothalamic-pituitary gonadal axis of male and female AR^{-/-} and AR^{+/+} chicken are needed to better understand how the steroidogenesis could be maintained while the gametogenesis is disrupted.

Further, since the cell lineages that give rise to the Sertoli and Leydig cells of the testis and the cell lineages that give rise to the theca and granulosa cells of the ovary differ between mammals and chickens [45], the regulatory mechanisms, including androgen-dependent ones, in these cell types involved in steroidogenesis and gametogenesis may differ between chickens and mammals. Thus, the expression of LHR and the number of Leydig cells in the mouse are AR-dependent [44, 46].

7. What is the reason behind androgen treatment to study the crowing behavior in the androgen receptor knockout, if the AR knockout animals are irresponsive to androgens. What happens if you don't induce the crowing with androgens? Do the knockout birds start crowing later than the -/+ or +/+ animals? Do they ever crow? The induction of crowing in roosters by testosterone is a well-known male-typical phenotype in chickens and landmark experiment in behavioral endocrinology. Here we show that this behaviour is indeed a typical androgenic response dependent on the androgen receptor, stating now in lines 283-287: In the testosterone-treated AR^{+/-} chicks, the development of a comb began at about 15 to 20 days of age, while testosterone-treated AR^{-/-} chicks did not develop a comb (Fig. 4), **thereby confirming the AR dependence of this sexual ornament.**

At adulthood, knockout roosters utter normal social calls but no male-typical crows but sometimes utter strange sounds. We mention now in line 278-281: **At 20 weeks, next to typical chicken social calls, adult AR^{-/-} roosters occasionally produce vocalizations that do not fit typical male or female patterns. These unique vocalizations shall be subject to detailed analysis in future studies.** The comprehensive validation of these vocalizations requires a detailed description of the entire vocal repertoire of adult hens and roosters, a task that is beyond the scope of this paper but will be published in the future.

8. The same applies to the testosterone treatment in the bursa and the comb growth. It would be more significant to show the actual differences between wild type and knockout. For example, a time course of the normal bursa regression in +/+ males compared with the -/- males. Supplementing the bursa data with apoptosis markers, like cleaved caspase 3, would strengthen the phenotype, making it consistent with the idea proposed. The bursa and the comb are widely recognized as testosterone-sensitive tissues in developing chickens, although some doubts persist. Our study confirms that these phenotypes indeed depend on androgen receptors, as previously mentioned. Regarding the bursa, we state now in lines 255-257: **Artificial induction of bursa involution in both sexes during the early embryonic stage may be due to testosterone treatment, although the exact underlying mechanisms are not clear and estrogen and progesterin activity are also thought to be involved [49].**

Providing a detailed timeline for bursa development is beyond the scope of this paper, as is the case for the comb development. Previous publications (e.g. previous Reference 48: Milicevic, Z., et al for the bursa) have covered the bursa timelines. Further, the timeline of the development of the size of the bursa in the knockouts is not very informative since it persists till adulthood in knockouts and almost completely degenerates in controls (former Fig. S6, now Fig. S7). Even without apoptosis marker, the comparison of age-matched histology shows much lower number of lymphoid follicles in testosterone-treated wild type chickens (former Fig. S6, now Fig. S7). The development of bursa cell types in the knockout, heterozygous and wildtype male and female will be published elsewhere.

Similarly, the comb does not develop at all in the knockouts and therefore we did not include a timeline

Minor comments:

9. Figure 2. Why you decided to run one-tailed t-test instead of ANOVA when more than one condition is present? Please provide a description of the analysis performed in these figures. Apologies for the oversight. As stated in the legend of Tab. S4 in the previous version, the statistical analysis was performed using “Standard Least Squares and incorporated Restricted Maximum Likelihood estimation, followed by LS Means Tukey HSD post hoc tests for pairwise comparisons”. We corrected this in the legend of Fig. 2 (lines 855-857).

10. In figure 1, it is stated that the scale bars represent figures in E panels, but there are only A-D panels. We corrected the mistake in line 816-817.

11. The Lacunar channels in figure S4 A to C seems to be a sectioning artefact rather than lacunar channels. We deleted the entire figure.

12. Discussion title is missing; Results sub-titles are missing. We included subtitles throughout.

New Methods: We updated the methods for FSFH and LH analysis (lines 696-713), the preparation of paraffin sections of the gonads (lines 721-728), the labelling of DMRT1 mRNA and LHR mRNA (lines 742-745) and lines (750-752), the measurement of the diameters of seminiferous tubules (lines 728-731), and the quantification of LHR mRNA represented by brownish dots per Leydig cell (lines 767-769).

REVIEWERS' COMMENTS

Reviewer #3 (Remarks to the Author):

The authors addressed all my comments and suggestions. They included new results to support or further characterize the described phenotypes. The manuscript is ready for publication.